# Association between vmPFC gray matter volume and smoking initiation in adolescents

Shitong Xiang [1,2,27], Tianye Jia [1,2,3,4,27] ✉, Chao Xie[1,2], Wei Cheng[1,2], Bader Chaarani[5], Tobias Banaschewski [6], Gareth J. Barker [7], Arun L. W. Bokde [8], Christian Büchel [9], Sylvane Desrivières [4], Herta Flor [10,11], Antoine Grigis[12], Penny A. Gowland [13], Rüdiger Brühl [14], Jean-Luc Martinot[15], Marie-Laure Paillère Martinot[15,16], Frauke Nees [6,10,17], Dimitri Papadopoulos Orfanos [11], Luise Poustka[18], Sarah Hohmann [6], Juliane H. Fröhner [19], Michael N. Smolka [19], Nilakshi Vaidya[20], Henrik Walter [20], Robert Whelan[21], Hugh Garavan[5], Gunter Schumann [1,3,20,22], Barbara J. Sahakian [1,23], Trevor W. Robbins [1,24] ✉, Jianfeng Feng [1,2,25,26] ✉ & IMAGEN Consortium*

Smoking of cigarettes among young adolescents is a pressing public health issue. However, the neural mechanisms underlying smoking initiation and sustenance during adolescence, especially the potential causal interactions between altered brain development and smoking behaviour, remain elusive. Here, using large longitudinal adolescence imaging genetic cohorts, we identify associations between left ventromedial prefrontal cortex (vmPFC) gray matter volume (GMV) and subsequent self-reported smoking initiation, and between right vmPFC GMV and the maintenance of smoking behaviour. Rule-breaking behaviour mediates the association between smaller left vmPFC GMV and smoking behaviour based on longitudinal cross-lagged analysis and Mendelian randomisation. In contrast, smoking behaviour associated longitudinal covariation of right vmPFC GMV and sensation seeking (especially hedonic experience) highlights a potential reward-based mechanism for sustaining addictive behaviour. Taken together, our findings reveal vmPFC GMV as a possible biomarker for the early stages of nicotine addiction, with implications for its prevention and treatment.

Nicotine intake, especially in the form of cigarette smoking, is the most prevalent addictive behaviour, and the leading cause of adult mortality around the world[1]. Approximately 1 in 5 deaths and $96.8 billion in productivity losses are attributable to smoking annually in the US[2], and yearly cigarette-smoking-related deaths are expected to reach 8 million worldwide by 2030[3]. Evidence suggests that cigarette smoking in childhood is associated with increased risk for psychiatric conditions and poorer cognitive function[4–6], potentially through its impact on brain development[7]. Smoking initiation is most likely to occur during adolescence, and previous studies in human cohort and animal models have suggested that early nicotine exposure during adolescence could directly increase the risk of nicotine dependence in the future[8–11]. It has also been observed that most daily smokers will develop nicotine dependence by age 18, whereas teenager non-smokers are unlikely

A full list of affiliations appears at the end of the paper. *A list of authors and their affiliations appears at the end of the paper.

✉ e-mail: tianyejia@fudan.edu.cn; twr2@cam.ac.uk; jianfeng64@gmail.com

ever to do so[12,13]. Further, quitting cigarette smoking is notoriously difficult once addiction has been established during adolescence[14,15]. There is a long latency from substance use to disorder (i.e., addiction), which offers a significant window of opportunity for clinical interventions. However, treatment efforts have focused almost exclusively on those with serious, usually chronic addictions, virtually ignoring the much larger population with pre-addiction[16]. Therefore, this tremendous burden on public health calls for further understanding of the biological mechanisms contributing to smoking initiation and early-stage sustenance.

The transition from adolescence to early adulthood is a period of critical brain development and maturation. While the growth curve of brain gray matter volume (GMV) peaks in preadolescence, brain maturational processes, including synaptic pruning and axon myelination, will continue throughout the entire period of adolescence[17]. Furthermore, these brain maturational processes are associated with brain re-organisation, a process underlying the maturation of cognitive control, thus improving adaptive behaviour, such as risky decision making and conduct control[12,18–20]. Several previous studies have attributed smoking initiation to impaired executive control and the underlying neural circuits[13,21]. For instance, the prefrontal cortex (PFC), the most critical neural network engaged in response inhibition and risk adjustment, continues to develop structurally and functionally into adulthood[20,22]. Its disrupted development has been implicated as a trigger for maladaptive behaviour, such as addiction[19,20,23]. On the other hand, substance use, including nicotine exposure, may cause damage to the brain and accelerate brain aging potentially through its neurotoxic properties, indirectly exacerbated by excessive smoking[8,10]. Such neurotoxic effects may also affect the reinforcement system itself and induce other forms of substance dependence[24,25]. Further, there are significant age differences in many of the acute neurobehavioural impacts of nicotine exposure[26]. For instance, early onset smokers exhibit deficits in reward processing and response inhibition, whereas other behavioural effects, such as physical craving for nicotine following withdrawal are greater in adults[26–29]. While several associations of cigarette smoking with cognitive function and brain structures have been established[30–32], no consensus has been reached on the exact causal relationship between brain development and smoking in adolescence and its underlying neurobehavioural mechanisms remain elusive.

Here, we examined the potential mutual causality of smoking and brain development with a large longitudinal, community-based sample of adolescents. Using a stratified approach, we first identified those brain regions differentiating cigarette smoking initiation at a future time and also those with altered development associated with continued smoking. We further investigated if distinct neurobehavioural circuits could represent the proposed mutual causal factors, respectively. Moreover, we substantiated the proposed causality through both cross-lagged longitudinal analysis and Mendelian randomisation (MR). This longitudinal study may thus provide a crucial insight into understanding the initiation and maintenance of addictive behaviour in adolescents, with further implications for substance abuse preventative interventions and treatment.

## Results

### Demographics of participants
The present study analysed participants enrolled in the IMAGEN project, a prospective, multicentre longitudinal imaging genetics study in 2000 healthy adolescents[33]. Participants' neuroimaging and behavioural data were assessed at ages 14 (baseline), 19 (the follow-up, FU) and 23 (the newly released follow-up data as validation, FU-age-23). Cigarette smoking for each time point (i.e., of age 14, 19, or 23) was measured by the item "How many occasions during your lifetime have you smoked cigarettes?" from the self-rated European School Survey Project on Alcohol and Other Drugs (ESPAD). Participants with scores greater than 0 were considered smokers at the interview. After quality

control, this study included 807 (36.4%, 444 female participants [55%]) participants with complete structural images and behavioural scores at both BL and FU. Among them, 181 participants, i.e., the baseline smokers (BL-S), smoked before the baseline interview. While about half of baseline smokers (87/181) had only once or twice experience of smoking before the baseline interview, most of them (166/181) did continue and report increased smoking frequency at the follow-up interview (Table 1). The remaining 626 participants were further subdivided into a control group ($n = 260$) and a follow-up smoker group (FU-S, $n = 366$) based on whether they reported ever smoking at the FU interview. Notably, while only 11 ( < 2%) and 58 ( ~ 7%) participants were reported daily smokers at baseline and follow-up, respectively, a majority of smokers (134/181 for BL-S and 280/366 for FU-S) did report smoking in the past 30 days before the follow-up interview, indicating an ongoing progress towards more regular nicotine use. The demographic characterisation found that smokers (i.e., both BL Smokers and FU Smokers) were more likely to take a risk and had worse risk adjustment compared with the Con Group ($|t| > 2.16$, Cohen's $d > 0.21$, $p < 0.030$). There were no significant differences in other basic characteristics (such as sex, handedness, BMI, IQ, mental health, and parental smoking exposure) between the smokers and controls, and more detailed demographic information can be found in Table 1.

### Smaller left vmPFC associated with smoking initiation
At baseline, compared with the controls, a whole brain analysis found BL-S with significantly smaller gray matter volumes (GMV) in clusters such as the ventromedial prefrontal cortex (vmPFC)/anterior cingulate cortex (ACC) (Peak Montreal Neurological Institute (MNI): [−2, 44, 6] Brodmann Area [BA] 10_L, $t_{426} = -4.46$, $p = 1E-5$, Cohen's $d = -0.44$, 95% Confidence Intervals (CI) = [−0.63, −0.25]; Cluster: 3962 voxels, $p_{FWE-adj} = 5E-7$), the left inferior frontal cortex (IFC, Peak MNI: [−50, 26, 27] BA 48_L, $t_{426} = -4.23$, $p = 3E-5$, Cohen's $d = -0.40$, 95% CI = [−0.61, −0.22]; Cluster: 446 voxels, $p_{FWE-adj} = 0.009$), the left lateral orbitofrontal cortex (latOFC, Peak MNI: [−36, 47, −3] BA 47_L, $t_{426} = -4.21$, $p = 3E-5$, Cohen's $d = -0.40$, 95% CI = [−0.61, −0.22]; Cluster: 775 voxels, $p_{FWE-adj} = 0.001$) and the right dorsolateral prefrontal cortex (dlPFC, Peak MNI: [39, 15, 27] BA 48_R, $t_{426} = -4.03$, $p = 7E-5$, Cohen's $d = -0.40$, 95% CI = [−0.59, −0.20]; Cluster: 796 voxels, $p_{FWE-adj} = 0.001$) (Fig. 1a upper). Remarkably, FU-S also had smaller GMV than the controls at baseline, i.e., prior to their smoking initiation, only in the left vmPFC from a whole-brain analysis (Peak MNI: [−5, 50, −5] BA 10_L, $t_{611} = -4.19$, $p = 3E-5$, Cohen's $d = -0.34$, 95% CI = [−0.50, −0.18]; Cluster: 438 voxels, $p_{FWE-adj} = 0.011$) (Fig. 1a lower), of which 426 voxels overlapped with the vmPFC cluster differentiating BL-S from the controls (Fig. 1b upper). We further verified the above between-group results within groups BL_S and FU_S, and again found that the GMV of left vmPFC (of the 426 overlapped voxels) at baseline was not only associated with the smoking frequency at baseline in BL-S ($r_{166} = -0.17$, $p_{one-tailed} = 0.016$, 95% CI = [−∞, −0.04], Fig. 1b upper) but could also predict the future smoking frequency in FU-S ($r_{351} = -0.11$, $p_{one-tailed} = 0.020$, 95% CI = [−∞, −0.02], Fig. 1b lower; also see Table S2). Also, both above associations remained significant after excluding occasional users, i.e., with once or twice experience life-time ($r_{79} = -0.21$, $p_{one-tailed} = 0.037$, 95% CI = [−∞, −0.02] in BL-S and $r_{261} = -0.11$, $p_{one-tailed} = 0.040$, 95% CI = [−∞, −0.01] in FU-S). Finally, lower GMV in the left vmPFC at baseline could additionally predict higher future smoking quantities within 30 days before the follow-up interview ($r_{166} = -0.16$, $p_{one-tailed} = 0.019$, 95% CI = [−∞, −0.03] in BL-S and $r_{351} = -0.09$, $p_{one-tailed} = 0.040$, 95% CI = [−∞, −0.01] in FU-S, Table S2). The above results hence indicate reduced GMV in the left vmPFC as a highly sensitive risk factor for the initiation of future smoking behaviour.

### Right vmPFC reduction from baseline associated with the maintenance of smoking
We next investigated if smoking was associated with the development of GMV (Fig. S1, see Methods for more details) and observed faster

## Table 1 | Demographic characteristics of the samples in this study

| Characteristics | All sample (n = 807) | Con group (n = 260) | BL-S Group (n = 181) | FU-S group (n = 366) |
|---|---|---|---|---|
| Sex (%) | | | | |
| Male | 363 (45%) | 106 (41%) | 84 (46%) | 173 (47%) |
| Female | 444 (55%) | 154 (59%) | 97 (54%) | 193 (53%) |
| Handedness (%) | | | | |
| Right | 717 (89%) | 232 (89%) | 166 (92%) | 319 (87%) |
| Left | 90 (11%) | 28 (11%) | 15 (8%) | 47 (13%) |
| Site (%) | | | | |
| London | 118 (15%) | 43 (17%) | 17 (9%)[a] | 58 (16%) |
| Nottingham | 109 (14%) | 39 (15%) | 24 (13%) | 46 (13%) |
| Dublin | 67 (8%) | 23 (9%) | 7 (4%) | 37 (10%) |
| Berlin | 68 (8%) | 21 (8%) | 20 (11%) | 27 (7%) |
| Hamburg | 116 (14%) | 37 (14%) | 26 (14%) | 53 (14%) |
| Mannheim | 98 (12%) | 26 (10%) | 36 (20%)[a] | 36 (10%) |
| Paris | 121 (15%) | 30 (12%) | 36 (20%)[a] | 55 (15%) |
| Dresden | 110 (14%) | 41 (16%) | 15 (8%)[a] | 54 (15%) |
| Family environment factors | | | | |
| Educational attainment (Mother) | 3.45 (1.75) | 3.46 (1.69) | 3.64 (1.77) | 3.35 (1.79) |
| Educational attainment (Father) | 3.39 (1.91) | 3.38 (1.85) | 3.61 (1.93) | 3.29 (1.93) |
| Parent smoking | 3.40 (2.58) | 3.16 (2.65) | 3.74 (2.42) | 3.39 (2.58) |
| Socioeconomic score | 0.61 (0.99) | 0.54 (0.92) | 0.74 (1.1)[a] | 0.59 (0.97) |
| Family stresses score | 2.63 (2.51) | 2.60 (2.55) | 2.96 (2.71) | 2.49 (2.37) |
| Negative life events | 6.36 (2.76) | 6.04 (2.64) | 7.27 (2.97)[a] | 6.07 (2.58) |
| Physical factors | | | | |
| Puberty score | 14.73 (2.72) | 14.68 (2.71) | 14.98 (2.55) | 14.63 (2.81) |
| Baseline TIV | 1415.67 (124.18) | 1413.02 (113.11) | 1399.42 (130.99) | 1425.58 (127.59) |
| Follow-up TIV | 1419.68 (131.70) | 1419.23 (124.09) | 1399.66 (135.4) | 1429.89 (134.27) |
| Baseline BMI | 20.48 (2.86) | 20.21 (2.77) | 20.61 (2.8) | 20.52 (2.94) |
| Follow-up BMI | 22.47 (3.32) | 22.25 (3.24) | 22.44 (3.22) | 22.65 (3.42) |
| Cognitive functions | | | | |
| WISCIV total score | 190.77 (22.08) | 191.90 (21.01) | 189.92 (21.87) | 190.39 (22.84) |
| AGN mean correct latency (Negative) | 498.51 (114.73) | 503.87 (106.67) | 494.17 (120.79) | 496.82 (116.25) |
| AGN mean correct latency (Positive) | 478.56 (108.68) | 480.97 (100.55) | 476.92 (117.99) | 477.52 (108.65) |
| AGN total omission number (Negative) | 12.07 (8.27) | 11.66 (8.29) | 13.13 (8.09) | 11.81 (8.32) |
| AGN total omission number (Positive) | 13.76 (7.62) | 13.46 (7.38) | 14.70 (7.88) | 13.46 (7.63) |
| CGT delay aversion | 0.24 (0.14) | 0.22 (0.13) | 0.26 (0.16) | 0.24 (0.14) |
| CGT Deliberation Time | 2080.88 (689.57) | 2045.99 (625.99) | 2018.58 (699.04) | 2095.21 (619.23) |
| CGT Overall Proportion Bet | 0.48 (0.13) | 0.47 (0.14) | 0.51 (0.13) | 0.47 (0.13) |
| CGT quality of decision making | 0.94 (0.09) | 0.94 (0.08) | 0.93 (0.09) | 0.94 (0.09) |
| CGT risk adjustment | 1.66 (0.94) | 1.8 (0.96) | 1.55 (0.91)[a] | 1.62 (0.95)[a] |
| CGT risk taking | 0.53 (0.14) | 0.49 (0.15) | 0.56 (0.14)[a] | 0.53 (0.14)[a] |
| PRM Total Correct Number | 95.23 (7.2) | 95.26 (7.26) | 95.18 (6.96) | 95.24 (7.3) |
| RVP Accuracy | 0.89 (0.05) | 0.89 (0.05) | 0.88 (0.04) | 0.89 (0.05) |
| SWM Between Errors | | 17.67 (13.01) | | 17.47 (13.36) |

## Table 1 (continued) | Demographic characteristics of the samples in this study

| Characteristics | All sample (n = 807) | Con group (n = 260) | BL-S Group (n = 181) | FU-S group (n = 366) |
|---|---|---|---|---|
| | | 18.04 (13.39) | 19.68 (13.92) | |
| SWM Strategy | 30.95 (5.52) | 31.32 (5.26) | 31.01 (5.49) | 30.85 (5.7) |
| Mental Disorders | | | | |
| ADHD | 37 (4.58%) | 12 (4.62%) | 8 (4.42%) | 17 (4.64%) |
| CD | 28 (3.47%) | 8 (3.08%) | 7 (3.87%) | 14 (3.83%) |
| ODD | 15 (1.86%) | 4 (1.54%) | 4 (2.21%) | 7 (1.91%) |
| GAD | 14 (1.73%) | 5 (1.92%) | 3 (1.66%) | 6 (1.64%) |
| MDD | 26 (3.22%) | 8 (3.08%) | 6 (3.31%) | 12 (3.28%) |
| Smoking characteristics | | | | |
| Age of first smoking | — | | 12.97 (0.97) | 16.08 (1.37) |
| Lifetime occasions of smoking at baseline | — | | 2.47 (1.78) | — |
| Lifetime occasions of smoking at follow-up | — | | 5.12 (1.52) | 3.61 (2.01) |
| Quantity of smoking in the last 30 days at baseline | — | | 0.57 (1.08) | — |
| Quantity of smoking in the last 30 days at follow-up | — | | 2.32 (1.87) | 1.20 (1.54) |

[a] means the variable differ significantly from the Con Group on the corresponding variable.
*TIV* Total intracranial volume, *BMI* Body mass index, *WISCIV* Wechsler intelligence scale for children – 4th Edition, *ADHD* attention deficit and hyperactivity disorder, *CD* conduct disorder, *ODD* oppositional defiant disorder, *GAD* generalised anxiety disorder, *MDD* major depression disorder, *AGN* affective go or no-go, *CGT* Cambridge gambling task, *PRM* pattern recognition memory, *RVP* rapid visual processing, *SWM* spatial working memory.

GMV reduction (i.e., from baseline to follow-up) in the right vmPFC only (Peak MNI: [10, 41, −11] BA 10_R, $t_{609} = -4.16$, $p = 4E-5$, Cohen's $d = -0.34$, 95% CI = [−0.50, −0.18]; Cluster: 747 voxels, $p_{FWE-adj} = 0.008$) when comparing FU-S to the controls with a whole brain analysis (Fig. 2a). Similar results were also observed between BL-S and the controls ($t_{424} = -2.04$, $p = 0.042$, Cohen's $d = -0.20$, 95% CI = [−0.39, −0.01], Table S2). It is notable that, at baseline, GMV in the right vmPFC showed no significant difference between FU-S and the controls ($t_{611} = -0.79$, $p = 0.747$, Cohen's $d = -0.05$, 95% CI = [−0.23, 0.10]), but was significantly smaller in the BL-S than the controls ($t_{426} = -1.97$, $p = 0.049$, Cohen's $d = -0.19$, 95% CI = [−0.39, 0], Table S1), thus indicating the altered GMV development might only occur after the initiation of smoking. We then verified the between-group findings within FU-S and confirmed the association of higher smoking frequency with reduced GMV in the right vmPFC ($r_{349} = -0.13$, $p_{one-tailed} = 0.010$, 95% CI = [-∞, −0.04], Fig. 2b). Again, this association remained significant after removing occasional users ($r_{259} = -0.14$, $p_{one-tailed} = 0.014$, 95% CI = [-∞, −0.03], Table S2). Additionally, reduced GMV in the right vmPFC was also associated with higher quantity of smoking in the last 30 days in FU-S ($r_{349} = -0.09$, $p_{one-tailed} = 0.026$, 95% CI = [-∞, −0.01], Table S2).

In contrast to these observations, frequency of smoking did not alter the development of the left vmPFC in either BL-S (compared to the controls: $t_{424} = 0.72$, $p = 0.470$, Cohen's $d = 0.07$, 95% CI = [−0.12, 0.26]) or FU-S (compared to the controls: $t_{609} = -1.00$, $p = 0.319$, Cohen's $d = -0.07$, 95% CI = [−0.24, 0.08], Table S1), which further supported a potential causal inference of smaller left vmPFC over higher smoking levels but not the other way around. Further, as baseline GMV was a strong indicator for future brain development of both left and right vmPFC ($r_{790} < -0.2$, $p < 0.001$), we reconducted the comparison of GMV changes in the right vmPFC between FU-S and the controls by matching their left and right vmPFC GMV (see Methods for more details) and again observed persistently greater reduced GMV in the right vmPFC of FU-S ($p < 0.05$ in 9984 out of 10000 resampling,

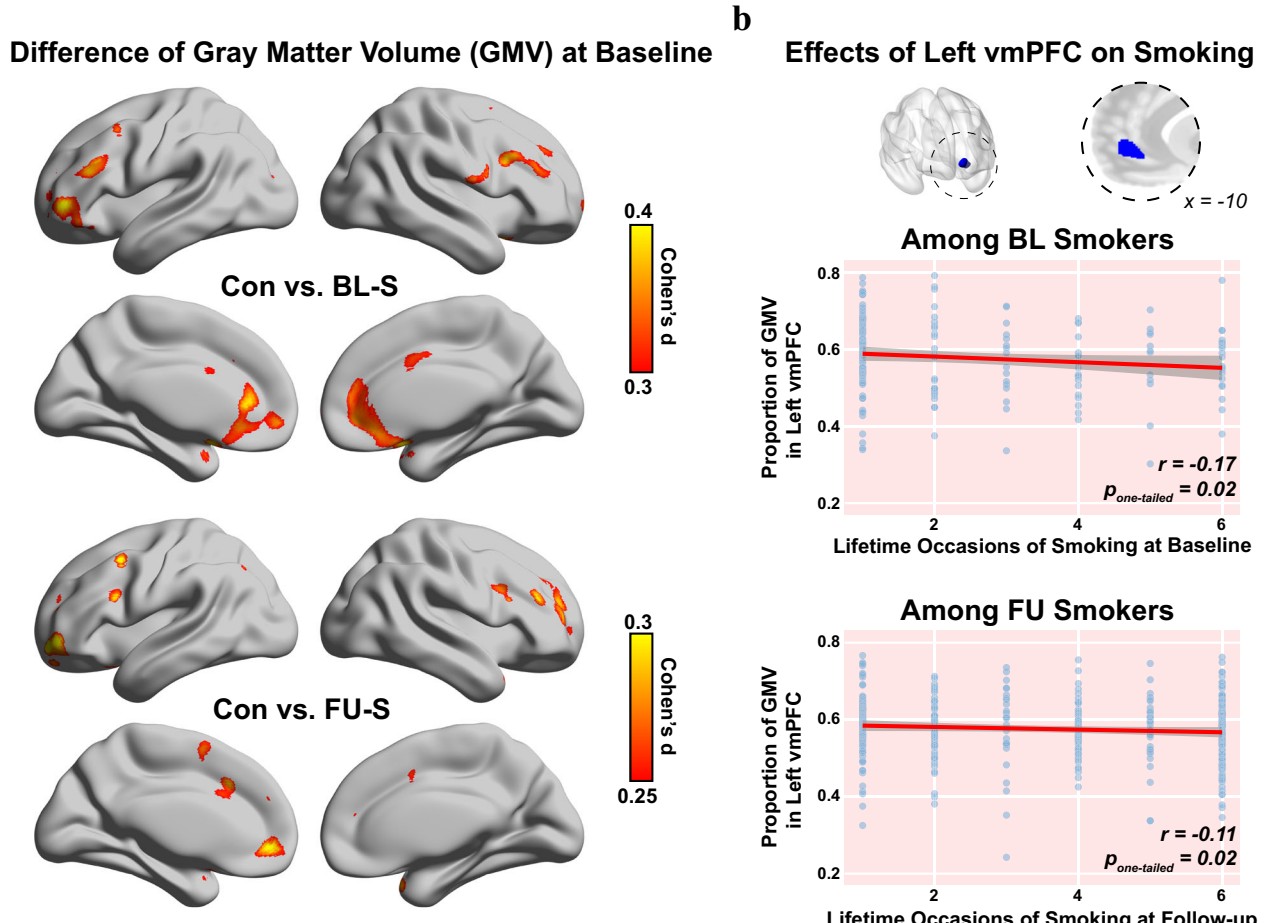

**Fig. 1 | Smaller Left vmPFC was associated with future smoking. a** Brain-wide difference of gray matter volume (GMV) at baseline between the controls (Con, $n = 260$) and smokers, i.e., baseline smokers (BL-S, $n = 181$) and follow-up smokers (FU-S, $n = 366$) through two-sample $t$-tests. **b** The correlations between GMV of the left vmPFC and quantity of smoking in BL-S (upper) and FU-S (lower). The one-tailed $p$-values were provided in the corresponding panels (also see Table S2). The error bands represented the 95% confidence intervals of the linear fitted models. Lifetime occasions of smoking were assessed with the question "How many occasions during your lifetime have you smoked cigarettes?" in European School Survey Project on Alcohol and Other Drugs (ESPAD, 1: '1-2'; 2: '3-5'; 3: '6-9'; 4: '10-19'; 5: '20-39'; 6: '40 or more'). Relevant source data were provided in the Source Data file.

$t_{mean} = −3.68$, $p_{mean} = 3E-4$, Cohen's $d = −0.37$, 95% = [−0.58, −0.18]). Additionally, in the newly released FU-age-23 data (4 years after FU), we observed a further (from FU to FU-age-23 greater reduction of GMV in the right vmPFC of group FU-S compared to the controls ($t_{430} = −2.39$, $p_{one-tailed} = 0.009$, Cohen's $d = −0.24$, 95% CI = [−∞, −0.07]), but again not for the left vmPFC ($t_{430} = −0.55$, $p_{one-tailed} = 0.291$, Cohen's $d = −0.06$, 95% = [−∞, 0.11], Fig. 2c). In conclusion, reduced GMV in the right vmPFC might be related to sustained smoking behaviour.

### Distinct roles of left and right vmPFC

To further characterise the neurobehavioural relevance of both left and right vmPFC, we investigated their associations with two personality traits, novelty seeking and sensation seeking, that have long been proposed to underlie the development of addictive behaviour[21,34]. Both higher novelty seeking (from TCI) and sensation seeking (from SURPS) were significantly associated with greater levels of smoking at BL (novelty seeking: $r_{792} = 0.19$, $p = 7E-8$, 95% CI = [0.12, 0.26]; sensation seeking: $r_{792} = 0.16$, $p = 6E-6$, 95% CI = [0.10, 0.23]) and FU (novelty seeking: $r_{792} = 0.35$, $p = 3E-24$, 95% CI = [0.29, 0.41]; sensation seeking: $r_{792} = 0.24$, $p = 7E-12$, 95% CI = [0.17, 0.30], Table S3).

However, while lower GMV in the left vmPFC at BL associated with higher novelty seeking at both BL and FU (BL: $r_{792} = −0.07$, $p = 0.040$, 95% CI = [−0.14, 0]; FU: $r_{792} = −0.10$, $p = 0.004$, 95% CI = [−0.17, −0.03]),

the association with sensation seeking was only significant at FU ($r_{792} = −0.09$, $p = 0.016$, 95% CI = [−0.16, −0.02], Table S3), which nevertheless reduced to non-significance after controlling for FU smoking, i.e., a complete mediation effect ($\beta_{mediation} = −2.03$, $p_{bootstrap} < 0.0001$; Fig. S2). Remarkably, this dedicated association between GMV of the left vmPFC and novelty seeking was only observed with the component 'disorderliness/rule-breaking' at BL ($r_{792} = −0.14$, $p = 8E-5$, 95% CI = [−0.21, −0.07], especially with sub-items "tci044" and "tci109"), i.e., the component most distinct from sensation seeking (significantly weaker than other components in novelty seeking; Steiger's test: $Z < −3.31$, $p < 0.001$, Cohen's $d = −0.14$, 95% CI = [−0.19, −0.05]; Table S4). On the other hand, the right vmPFC had an exclusive association with sensation seeking but not novelty seeking, where greater reduction of GMV (i.e., from BL to FU) in the right vmPFC demonstrated a significantly stronger association with higher sensation seeking ($r_{790} = −0.13$, $p = 2E-4$, 95% CI = [−0.20, −0.06]) than that with novelty seeking at FU ($r_{790} = −0.04$, $p = 0.261$, 95% CI = [−0.11, 0.03]; Steiger's test: $Z = −2.12$, $p = 0.03$, Cohen's $d = −0.08$, 95% CI = [−0.15, −0.01]; Table S3). Therefore, we demonstrated distinct neurobehavioural associations of GMV in the left vmPFC with disorderliness/rule-breaking on a questionnaire of novelty seeking, while in contrast, associations of GMV in the right vmPFC were specific to questions concerning sensation seeking.

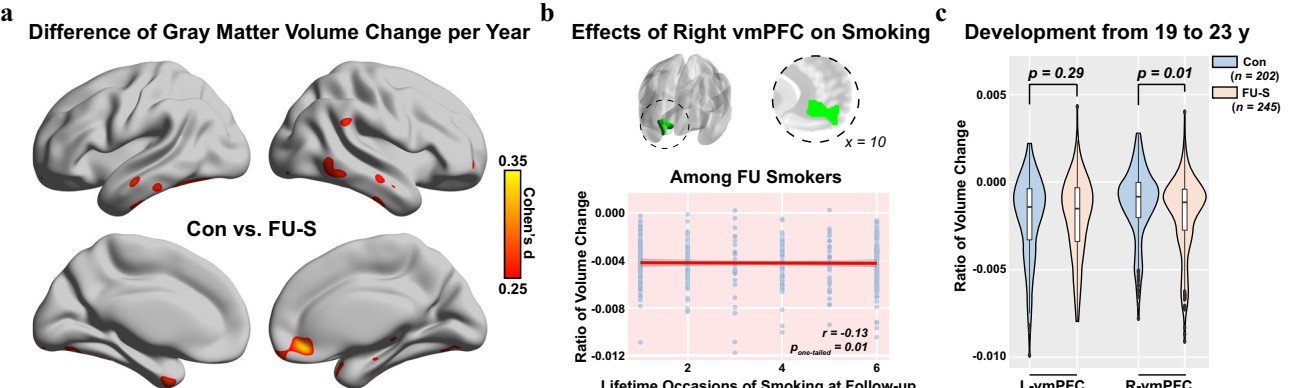

**Fig. 2 | Smoking was associated with faster reduction of gray matter volume (GMV) in the right vmPFC. a** Brain-wide difference of the development of GMV from the period of baseline to 5-year later follow-up between the controls (Con, $n = 260$) and follow-up smokers (FU-S, $n = 366$) through a two-sample $t$-test. **b** The correlations between the reduction of GMV in the right vmPFC and the quantity of smoking among FU-S. The one-tailed $p$-values were provided in the corresponding panels (also see Table S2). The error bands represented the 95% confidence intervals of the linear fitted models. **c** The difference in the development of GMV in bilateral vmPFC from the period of first follow-up to 4-year later second follow-up between the Con ($n = 202$) and FU-S ($n = 245$) through two sample $t$-tests. The upper and lower boundaries of each boxplot represented the first (Q1) and third (Q3) quantiles, respectively. Hence, the box body covered 50% of the central data (Inter Quartile Range, IQR), with the median marked by a central line. The top/bottom whiskers represented the maximun or minimum, respectivley without outliers. The outliers were identified as greater than Q3 + 1.5*IQR or less than Q1-1.5*IQR in the data distribution. Relevant source data were provided in the Source Data file.

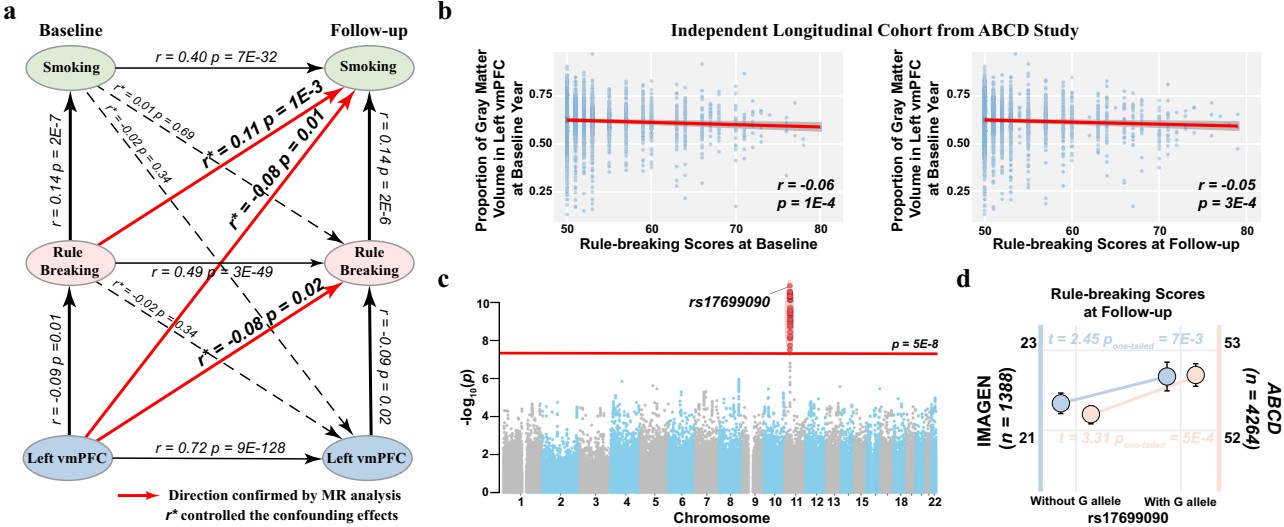

**Fig. 3 | The neurobehavioural circuit from the left vmPFC to rule-breaking serves as an inheritable risk factor in initiating addictive behaviour. a** The associations between gray matter volume (GMV) of the left vmPFC, rule-breaking scores and smoking were investigated by longitudinal cross-lagged analysis and causality explored with Mendelian randomisation (MR). Solid lines represented significant associations, and red lines represented causal inference confirmed with MR. Also, see Table S5. The two-tailed $p$-values obtained in a longitudinal cross-lagged analysis were provided. **b** The correlations between GMV of the left vmPFC and rule-breaking scores of children (at ages 9-10) from an independent longitudinal cohort ABCD. The two-tailed $p$-values were provided in the corresponding panels. The error bands represented the 95% confidence intervals of the linear fitted models. **c** The Manhattan plot of genome-wide associations for GMV of the left vmPFC. The red line indicated the genome-wide significance level (i.e., $p = 5E-8$). Variants with significant association were highlighted with red dots. **d** Individuals with the G-allele of rs17699090 had higher rule-breaking scores in both IMAGEN ($n = 1388$) and ABCD ($n = 4264$) cohorts. Two-sample $t$-tests were preformed in the comparision between the individuals with and without the G-allele of rs17699090. The one-tailed $p$-values were provided in the corresponding panel. The circles represented the average scores and the error bars represented the standad errors of rule-breaking scores. Relevant source data were provided in the Source Data file.

## Longitudinal and Mendelian randomisation evidence of L-vmPFC association with rule-breaking

With a cross-lagged longitudinal analysis, lower GMV in the left vmPFC at BL could predict future higher rule-breaking scores ($r = -0.08$, $p = 0.020$, 95% CI = [−0.15, −0.01]) and higher smoking ($r = -0.08$, $p = 0.012$, 95% CI = [−0.15, −0.01]) at FU, controlling the confounding effects at BL and FU (Fig. 3a). Similarly, higher rule-breaking scores at BL could predict future higher smoking at FU ($r = 0.11$, $p = 0.001$, 95% CI = [−0.17, −0.04], Fig. 3a). On the other hand, the reversed predictions did not hold (Fig. 3a). Thus, we can infer a potential causal chain

leading to smoking: "left vmPFC GMV->rule-breaking->smoking". To further confirm this chain of causal inference, we implemented a modified Mendelian randomisation (MR) analysis to investigate potential causal effects by introducing non-pleiotropic polygenic risk scores (PRS) as instrument variables, i.e., the valid-PRS (see Methods for more details)[35,36]. We first conducted GWAS for GMV in the left vmPFC, rule-breaking scores and quantity of smoking at BL in the remaining subjects from the IMAGEN project ($n = 1026$), who were not included in the above longitudinal analyses due to the lack of neuroimaging data at FU (see Methods for more details). We then calculated

the PRSs of above phenotypes in the independent longitudinal sample, and these PRSs all showed consistent positive associations with the corresponding phenotypes (Table S5). We observed consistent negative associations of valid-PRS$_{left-vmPFC}$ with rule-breaking scores and the quantity of smoking as well as positive associations of valid-PRS$_{rule-breaking}$ with the quantity of smoking as expected. On the contrary, the reversed prediction did not stand (Table S5). Hence, we could summarise a probable causal inference from the GMV in the left vmPFC through rule-breaking to smoking, but not vice versa. Further analyses also revealed that smaller GMV of the left vmPFC and the induced higher rule-breaking scores preceded future conduct problems as well as the other substance use (i.e., alcohol and marijuana) (longitudinal analysis: $|r| > 0.07$, $p < 0.05$; MR analysis: $|r_{737}| > 0.06$, $p_{one-tailed} < 0.04$; Fig. S3a–c & Table S6), thus highlighting the left vmPFC's potential causal involvement in rule-breaking behaviour. It is also notable that there was no association of conduct problems with the development of GMV in the right vmPFC ($r_{790} = −0.03$, $p = 0.396$, 95% CI = [−0.10, 0.04]) and sensation seeking (BL: $r_{792} = −0.02$, $p = 0.621$, 95% CI = [−0.09, 0.05]; FU: $r_{792} = 0.04$, $p = 0.266$, 95% CI = [−0.03, 0.11]), thus again indicating the distinct roles of left and right vmPFC. Importantly, in a very large independent longitudinal cohort of pre-adolescence (at ages 9–10) from the Adolescent Brain Cognitive Development[37] study ($N = 4415$), we again observed the negative associations of GMV in the left vmPFC at BL with rule-breaking scores (from the Child Behaviour Checklist) at both BL and FU ($r_{4388} < −0.05$, $p < 0.001$; Fig. 3b). Also, by applying the GWAS results obtained from IMAGEN, we verified the corresponding PRSs of both GMV in the left vmPFC ($r_{max} = 0.16$, $p_{one-tailed} = 3E-27$, 95% CI = [0.14, ∞]) and rule-breaking behaviour ($r_{max} = 0.04$, $p_{one-tailed} = 0.006$, 95% CI = [0.02, ∞]) in the ABCD cohort (Table S7). The PRS findings hence indicate that similar genetic constructs exist for both cohorts. Further, MR analyses based on valid-PRS again reached a conclusion on the causality of GMV in the left vmPFC over rule-breaking behaviour ($r_{4365} < −0.025$, $p_{one-tailed} < 0.050$, Table S7) in the ABCD cohort, where the effect sizes are roughly half of those in the imaging-behavioral associations ($r < −0.05$) similar to the observations from the IMAGEN cohort (i.e., from $r ∼ −0.14$ to $r ∼ −0.07$), thus suggesting this potential causal effect persisting throughout the whole adolescent developmental stage.

To further investigate whether this shared neurobehavioural circuit has underlying genetic factors, we performed a meta-analysis of GWAS for the GMV of left vmPFC in both IMAGEN ($N = 1778$) and ABCD ($N = 4390$) cohorts (see Methods for more details, Fig. S4), and identified a genome-wide significant QTL with the lead SNP rs17699090 (for the minor G-allele: meta-analysis, $Z$-score = 6.77, $p = 1.3E-11$, Cohen's $d = 0.09$, 95% CI = [0.06, 0.11]; ABCD, $r_{4342} = −0.08$, $p = 1.6E-7$, 95% CI = [−0.11, −0.05]; IMAGEN, $r_{1673} = −0.11$, $p = 1.2E-5$, 95% CI = [−0.16, −0.06]; Fig. 3c & Fig. S5; also see Fig. S6 for the forest plots for each cohort). SNP rs17699090 is within the *CAPRIN1* gene encoding *Caprin1* protein, which regulates the transport and translation of mRNAs for proteins involved in synaptic plasticity and neuron maturation[38], and *RNG105* (*Caprin1*) deficit mice will demonstrate abnormal development[39]. Additionally, complying with the proposed causal effects of lower GMV in the left vmPFC identified above, the participants with the G-allele of rs17699090 have higher rule-breaking score at follow-up interview than those without the G-allele in both the IMAGEN ($t_{1361} = 2.45$, $p_{one-tailed} = 0.007$, Cohen's $d = 0.13$, 95% CI = [0.04, ∞]) and ABCD cohorts ($t_{4216} = 3.31$, $p_{one-tailed} = 5E-4$, Cohen's $d = 0.10$, 95% CI = [0.05, ∞], Fig. 3d). Further, smokers in the IMAGEN cohort did carry more G-alleles of rs17699090 than those in controls ($t_{725} = 2.12$, $p_{one-tailed} = 0.018$, Cohen's $d = 0.12$, 95% CI = [0.04, ∞]). Notably, evidence also indicated that the genetic factors of GMV in the left vmPFC might be time sensitive for a specific development period adolescence, as the corresponding PRS demonstrated a series of rapidly diminished predictive effects across cohorts with increased age gaps deviated from age 14 (i.e., IMAGEN-BL, age 14; ABCD, ages

9–10; IMAGEN-FU, age 19; IMAGEN-FU2, age 23; HCP, ages 22–37; UKB1, ages 40–50; UKB2, ages 50–60; UKB3, ages > 60 years; Spearman's ranked test: $ρ = −0.976$, $p = 2E-7$, 95% CI = [−0.99, −0.87]; $r = 0.985$, $p = 2E-7$, 95% CI = [−0.99, −0.92] for a log-regression model; Table S8 and Fig. S7). The current results thus indicated that reduced GMV in the left vmPFC could be an inheritable factor of rule-breaking behaviour, such as conduct problems and early substance use, especially in pre-adult.

## R-vmPFC GMV association with reward reinforcement

Similar to the moderation effect of smoking over reduced GMV in the right vmPFC (Fig. 2), the longitudinal increase (from BL to FU) of sensation seeking was also significantly strengthened with past smoking experience (smokers vs controls: $t_{792} = 4.49$, $p = 8E-6$, Cohen's $d = 0.31$, 95% CI = [0.18, 0.46], Fig. 4a). Remarkably, the longitudinal covariation between increased sensation seeking and reduced right vmPFC ($r_{790} = −0.08$, $p = 0.037$, 95% CI = [−0.15, −0.01], Fig. 4b) was again strengthened by smoking (FU-S vs the controls: $Z = −2.55$, $p_{one-tailed} = 0.005$, Cohen's $d = −0.21$, 95% CI = [−∞, −0.06]; BL-S vs the controls: $Z = −1.76$, $p_{one-tailed} = 0.04$, Cohen's $d = −0.17$, 95% CI = [−∞, −0.01]; Fig. 4c). Importantly, only subitems that highlight the hedonic experience (i.e., surps6 'I enjoy new and exciting experiences even if they are unconventional' and surps9 'I like doing things that frighten me a little'), but not those simply seeking a novel or exotic experience (for instance, surps12 'I would like to learn how to drive a motorcycle.'), have their longitudinal changes associated with the development of the right vmPFC (Table S9). Again, the negative longitudinal covariation between surps6 and GMV in the right vmPFC was found to be much stronger in individuals with smoking experience (BL-S vs the controls: $Z = −1.86$, $p_{one-tailed} = 0.03$, Cohen's $d = −0.18$, 95% CI = [−∞, −0.02]; FU-S vs the controls: $Z = −3.23$, $p_{one-tailed} < 0.001$, Cohen's $d = −0.26$, 95% CI = [−∞, −0.13], Fig. 4d). Hence, the above results indicated that the right vmPFC GMV might serve as a general neural basis underlying the maintenance of pleasure (i.e., hedonic motivation), thus reinforcing substance use. Indeed, greater reduction of GMV in the right vmPFC was also associated with increased marijuana use ($r_{790} = −0.09$, $p = 0.016$, 95% CI = [−0.16, −0.02], Fig. 4e) and binge drinking ($r_{790} = −0.18$, $p = 2E-7$, 95% CI = [−0.25, −0.11], Fig. 4f).

Finally, while we cannot directly assess the impact of our neural biomarkers on addictive symptoms due to the minimal sample size of daily smokers in IMAGEN, we generalised our findings and verified the distinct role of both left and right vmPFC GMV in 330 regular/daily smokers from the Human Connectome Project[40]. To elaborate, smaller left vmPFC was only associated ($r_{323} = 0.15$, $p_{one-tailed} = 0.005$, 95% CI = [0.06, ∞]) with earlier initiation age of smoking, stronger than that with the right vmPFC (Steiger's $Z = 1.989$, $p_{one-tailed} = 0.047$, Cohen's $d = 0.11$, 95% CI = [0.04, ∞]). In contrast, smaller right vmPFC was associated with higher nicotine dependence (i.e., the FTND total score, $r_{323} = −0.14$, $p_{one-tailed} = 0.012$, 95% CI = [−∞, −0.05]; and DSM tobacco withdrawal symptoms $r_{323} = −0.12$, $p_{one-tailed} = 0.025$, 95% CI = [−∞, −0.03]), and both associations were stronger than the left vmPFC (Steiger's $Z = −2.243$, $p_{one-tailed} = 0.025$, Cohen's $d = −0.12$, 95% CI = [∞, −0.03] and Steiger's $Z = −3.184$, $p_{one-tailed} = 0.001$, Cohen's $d = −0.18$, 95% CI = [−∞, −0.09] respectively) (Table S10). The above findings were highly consistent with the proposed initiation role of the left vmPFC and reinforcement role of the right vmPFC in this study.

## Discussion

Using the largest longitudinal neuroimaging genetic study from adolescence to early adulthood, we revealed the prominent but distinct roles of left and right vmPFC in initiating and sustaining smoking and other potentially addictive behaviour. Specifically, reduced GMV in the left vmPFC was associated with increased rule-breaking, which could well extend to the violation of social (e.g. parental, school) rules about

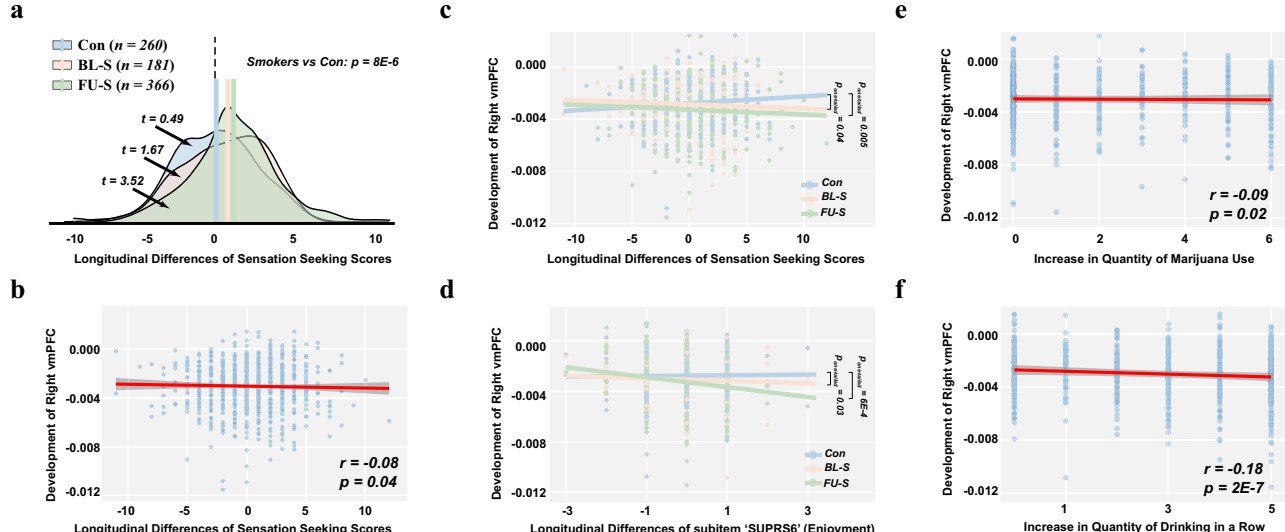

**Fig. 4 | The right vmPFC gray matter volume (GMV) association with sensation seeking and substance use scores. a** The distributions of longitudinal changes in sensation seeking scores of the three groups. One-sample t-tests were used to estimate the longitudinal differences of sensation seeking scores within each group. A two-sample t-tset was performed in the comparision between the smokers (i.e., both baseline smokers (BL-S) and follow-up smokers (FU-S)) and controls (Con), and the two-tailed p-value was provided. **b, e, f** The correlations between the development of gray matter volume (GMV) in the right vmPFC (from the period of baseline to 5-year later follow-up) and the increase of sensation seeking scores, the

quantity of marijuana use and quantity of binge drinking, respectively. The two-tailed p-values were provided in the corresponding panels. The error bands represented the 95% confidence intervals of the linear fitted models. **c, d** The correlations between the development of GMV in the right vmPFC and the increase of sensation seeking scores, as well as the subitem' SUPRS6', in each of the three groups. The group differences of these correlations (i.e. BL-S vs Con and FU-S vs Con) were evaluated with Steiger's Z tests, and one-tailed p-values were provided in the corresponding panels. Relevant source data were provided in the Source Data file.

smoking, possibly because the loss of function in the left vmPFC leads to behavioural disinhibition due to the discounting of consequences of rule-breaking[41]. Following this initiation of smoking, the reduced GMV in the right vmPFC may subsequently sustain and thus strengthen smoking behaviour further by removing inhibitory constraints on reward seeking and heightening the hedonic experience of smoking. vmPFC, a key node in the cortico-mesolimbic dopaminergic system, has long been proposed to regulate various aspects of cognitive function, such as risk adjustment and response inhibition[30,42,43]. Such a regulatory model is thus consistent with a general neural model of lateralisation of the regulation of effects on punishment and reward in the vmPFC[44–46].

Notably, while environmental factors, such as parents' smoking and pregnancy smoking[47], may also contribute to smoking during adolescents in the IMAGEN cohort ($r_{792} > 0.11$, $p < 0.01$), they were not associated with GMV in the left vmPFC at baseline ($|r_{792}| < 0.02$, $p > 0.57$), thus indicating independent contributions. Therefore, adolescents with smaller left vmPFC may need extra social support to reduce their exposure to risky environments. Providing alternative non-drug rewards at the early stage of substance use may help prevent the transition to substance dependency, as suggested by recent studies[48,49].

Compared with rule-breaking behaviour, sensation-seeking peaks later during early adulthood and requires accumulation of hedonic experience[34,50] leading to reinforcement of addictive behaviour. This may help to explain the differentiated associations of sensation seeking with substance use[51] and (more weakly) conduct problems in that while higher rule-breaking behaviour could underlie both substance use and conduct problems, only substance use could accumulate hedonic experience and hence reinforce increased sensation seeking. As sensation seeking has long been implicated in the reinforcing effects of psychostimulants[52], we thus proposed a reinforcement loop underlying the maintenance of smoking, where smoking behaviour could reduce GMV in the right vmPFC, leading to disinhibited reward-seeking behaviour (i.e., higher sensation seeking), and hence, in turn,

promote future smoking behaviour. This hypothesis was further supported by the distinct association of the right vmPFC (significantly stronger than the non-significant left vmPFC) GMV with nicotine dependence in daily smokers from the HCP data. Therefore, psychotropic drugs or therapies that preserve GMV in the right vmPFC or enhance its function, for instance, using rTMS[53], could be potential treatments for addiction.

It is a limitation that the current study mainly focused on the initiation and the early stage of addictive behaviour, but the onset of actual addiction requires possibly aberrant reinforcement processes operating long-term[54]. Nevertheless, we did manage to validate the proposed differentiated roles of the left and right vmPFC GMV in addictive behaviour (i.e., with initiation and sustenance/dependence respectively) in daily smokers from the independent HCP data. In future studies, it would be of considerable interest to understand how sustained smoking behaviour, driven initially by hedonic experience may further develop into dependence (i.e., psychological or physical craving for nicotine following withdraw), where habit-inducing regions such as the insula and striatum might eventually become involved[55,56]. Therefore, longitudinal data of patients with substance use disorder and relevant animal models are crucial for the future studies. The other limitation of this study was that some of the genetic-behavioural findings in the generalisation cohort were relatively small, which could be largely due to the fact that the genetic constructs of left vmPFC GMV were highly time sensitive (Table S8 and Fig. S7). Furthermore, in the context of the vaping epidemic among adolescents, whether there are similar neurobehavioural mechanisms between neurodevelopment and e-cigarette use needs further investigation.

In summary, we identified distinct neural bases for the initiation and sustenance of substance use, represented by reduced GMV in the left and right vmPFC, respectively. This reduction in GMV in the left vmPFC has a possible causal influence on rule-breaking behaviour that potentially leads to the initiation of substance use. Complementarily, the substance-induced changes in GMV in the right vmPFC may modulate hedonic effects of substance use, which, in turn, reinforces

and maintains future substance use. Thus, our findings provide a possible causal account of how smoking is initiated and then sustained, potentially leading to dependence.

## Methods

### Participants

The dataset used for the present study was selected from the IMAGEN project, which was a prospective, multicentre longitudinal imaging genetics study that recruited 2000 healthy adolescents. The standard operating procedures for the IMAGEN project are available at https://imagen-project.org/, which contain details on ethics, recruitment, neuropsychological tests and scanning protocols of magnetic resonance imaging (MRI) data[33,57].

Characteristic information is assessed using multiple questionnaires, assessments and tasks. In brief, socioeconomic and family stress scores were rated according to the Development and Well-being Assessment (DAWBA, parent-rated) family stresses total score and socioeconomic item, with greater scores indicating poorer family environments. The Life Events Questionnaire (LEQ) assesses positive and negative life events in childhood and young adulthood. Negative life events scores were computed as the sum of the frequencies of 20 negative experiences as suggested by previous study[58]. The puberty scores were rated according to the Pubertal Development Scale (PDS, self-rated). IQ of each individual was computed as the total score of the Wechsler Intelligence Scale for Children - Fourth Edition (WISC-IV). Other cognitive functions were measured using the Cambridge Neuropsychological Test Automated Battery (CANTAB, http://www.cambridgecognition.com), which comprises tasks of Affective Go/No-go (AGN), Cambridge Guessing Task (CGT), Pattern Recognition Memory (PRM) (total correct number), Rapid Visual Processing (RVP) and Spatial Working Memory (SWM). Mental health stages were assessed with DAWBA (parent-rated) at baseline interview. According to the Diagnostic and Statistical Manual of Mental Disorders (DSM), the computer prediction scores for comment mental disorders (including Attention Deficit and Hyperactivity Disorder, Conduct Disorder, Oppositional Defiant Disorder, Generalised Anxiety and Major Depression) were rated, and participants with diagnostic risk greater than 50% were reported as cases. All above demographic information was summarised in Table 1. Accordingly, significant characteristic information, i.e., sex, handedness, research sites, BMI, total intracranial volume (TIV), socioeconomic score, negative life events and IQ, were regressed out as covariates in the following analyses.

### Measurement of substance use

Cigarette smoking for each time point (i.e., of age 14, 19 or 23) was measured by the item "How many occasions during your lifetime have you smoked cigarettes?" from the European School Survey Project on Alcohol and Other Drugs (ESPAD). Participants with scores greater than 0 were considered smokers at the interview. For instance, participants with smoking experience at the baseline interview (14 years old) were assigned to the baseline smokers group (BL-S). The remaining participants were further divided into a control group (Con) and a follow-up smokers group (FU-S) according to whether they reported smoking experience at the follow-up interview (19 years old).

Additionally, drinking and Marijuana use were measured with the item "How many occasions in your whole lifetime have you had any alcoholic beverage to drink?" and the item "How many occasions in your whole lifetime have you used marijuana (grass, pot) or hashish (hash, hash oil)?" from the European School Survey Project on Alcohol and Other Drugs (ESPAD)[59], respectively.

### MRI data acquisition and processing

MRI data were acquired at eight IMAGEN assessment sites with 3 T MRI scanners of different manufacturers (Siemens, Philips, General Electric, Bruker). The scanning variables were specifically chosen to be compatible with all scanners. In addition, imaging protocols were harmonised across sites and scanners. Based on the ADNI protocol, high-resolution 3-dimensional T1-weighted images (1.1 mm isotropic voxel size) were acquired with a gradient-echo sequence (http://adni.loni.usc.edu/methods/mri-analysis/mri-acquisition).

Processing of the structural T1-weighted images was performed with the Statistical Parametric Mapping toolbox (SPM, Wellcome Department of Neuroimaging, London, United Kingdom; http://www.fil.ion.ucl.ac.uk/spm) implemented by Matlab (R2020b). For voxel-based morphometry (VBM), the images were first segmented into gray matter, white matter and cerebrospinal fluid tissue maps with the VBM8 toolbox (http://www.neuro.uni-jena.de/vbm). Then the gray matter maps were used to generate the study-specific template by the DARTEL algebra[60]. Next, the gray matter maps were warped to standard Montreal Neurological Institute (MNI) space ($1.5 \times 1.5 \times 1.5$ mm resolution) with an iterative registration and modulated by multiplying the linear and nonlinear components of the Jacobian determinants. Finally, the gray matter maps were smoothed with a full width at the FWHM Gaussian kernel of 8 mm, and the value of each voxel was the proportion of gray matter volume (GMV). For gray matter development[25], both the BL and FU scans from each participant were first processed with the pairwise longitudinal tool in SPM to generate a within-subject average template and the corresponding Jacobian determinants maps. Then the within-subject average templates were segmented and warped to the standard MNI space with the same pipeline as VBM. Next, the native space gray matter maps from the within-subject average template were multiplied voxel-by-voxel with the corresponding Jacobian determinants maps to obtain the gray matter volumetric changes rate maps. The transform fields were then applied to the gray matter volumetric change rate maps to normalise it to standard MNI space. Finally, the gray matter volumetric changes rate maps were also smoothed with a full width at an FWHM Gaussian kernel of 8 mm. The value of each voxel represented the voxel-wise volumetric change rate per year. Voxel-wise comparisons between the controls and smokers (BL-S or FU-S) were obtained with two-sample t-tests. Hence, significant clusters after familywise error adjustment were identified using random field theory as implemented in SPM.

### Resampling data

To confirm that reduced GMV in the right vmPFC was indeed induced by early smoking experience rather than a by-product of smaller vmPFC at baseline, we randomly resampled 200 individuals from the Con and FU-S, respectively, with equivalent GMV between the two groups in both left and right vmPFC at baseline interview. We repeated the above resampling 10000 times and compared the GMV trajectories of the right vmPFC between the Con and FU-S groups at each resampling.

### Measurement of personality

In the present study, two personality traits, i.e., novelty seeking and sensation seeking, were assessed. The novelty seeking score was rated according to the Temperament and Character Inventory-Revised (TCI-R, self-rated)[61], which included four subscale components, i.e., "Disorderliness/Rule-breaking", "Impulsiveness", "Extravagance", and "Exploratory Excitability". The sensation seeking score was rated according to the Substance Use Risk Personality Scale (SURPS, self-rated)[62]. Notably, while both novelty and sensation seeking are characterised by the pro-active pursuit of thrilling, exotic or hedonic experience[62,63], they target distinct behavioural aspects. For instance, while sensation seeking includes subitems highlighting the enjoyment of previous sensational experiences (i.e., hedonic experience), novelty seeking is featured with response disinhibition components, i.e., disorderliness/rule-breaking and impulsiveness, in addition to exploratory excitability and extravagance components that are also partly covered by sensation seeking.

## Cross-lagged longitudinal analysis

The longitudinal association between phenotypes was explored using a classic two-wave cross-lagged panel model (implemented with the lavaan package version 0.8 in R)[64]. In brief, covariates were regressed out before the analysis. Model parameters were estimated by maximum likelihood estimation. Standardised regression coefficients and corresponding P values were reported.

## Mendelian randomisation (MR) analysis

In the present study, a modified Mendelian randomisation approach was performed to investigate a potential causal relationship[36,65], in which non-pleiotropic polygenic risk scores (i.e., the valid-PRS) were established as instrument variables of randomised experiments for potential causal inference. A valid instrument variable should only affect the explanatory variable, but not the outcome variables if not through the explanatory variables[35].

Specifically, using PLINK[66], we conducted exploratory GWAS for the phenotypes of interest in the leftover participants excluded from the above analyses due to the lack of neuroimaging information at FU ($n = 1026$) in the IMAGEN project. We first performed quality-control processing using PLINK[66], where SNPs with call rates <95%, minor allele frequency <0.1%, deviation from the Hardy–Weinberg equilibrium with $p < 1E-10$ were excluded from the analysis. Then we conducted the imputation on the quality-controlled genetic data with the TOPMed imputation server (https://imputation.biodatacatalyst.nhlbi.nih.gov). After imputation, 5966316 SNPs were available for IMAGEN sample. The following GWAS analysis was performed with sex, research sites and top 10 ancestry principal components as covariates (see Fig. S8 for the plot of the first two PCs projected to 1000 Genome data). P-value-informed clumping with a cutoff at $r^2 = 0.1$ in a 250 kb window was then implemented to minimise SNP overrepresentation while maintaining the most informative signals across the genome. PRSs were hence calculated with clumped SNPs for individuals with complete longitudinal neuroimaging, behaviour and genetic information ($n = 752$) at a pre-defined set of P-value thresholds (i.e., 0.05, 0.1, 0.2, 0.3, 0.4, and 0.5). Please be noted that the highest co-ancestry level (i.e., the proportion of shared genome estimated using PLINK 2.0) in the IMAGEN cohort is much lower than the common threshold 0.125, i.e., third-degree relatives, for exclusion (Fig. S9), so the risk for information leakage due to relatedness between the discovery GWAS sample and the target PRS sample is negligible.

The threshold of each PRS being most significantly associated with the corresponding explanatory variable (at both baseline and follow-up interview) was selected to establish validated instrumental variables in the following steps. Next, we step-wisely removed potential pleiotropic SNPs from the above PRSs to obtain the valid-PRSs, again based on a pre-defined set of P-value thresholds for the exploratory GWAS (i.e., 0.05, 0.1, 0.2, 0.3, 0.4, and 0.5). For instance, at the threshold of P-value = 0.50, any SNPs having P-values < 0.50 with both phenotypes of interest will be removed from the calculation of PRSs, hence keeping only SNPs dedicated for one phenotype but with less than random effects on the other. A valid-PRS established based on such a process thus fulfils the definition of an instrumental variable mentioned above if it remains associated with the designated phenotype of interest. Finally, cross associations between the valid-PRSs of the explanatory variables and the phenotypes of the corresponding outcome variables were investigated. A significant cross-association would establish a potential causal inference from the explanatory variables to the outcome variables according to the argument of randomised experiments.

## Adolescent Brain Cognitive Development study

An independent longitudinal genetic-imaging cohort, the ABCD study (acquired from Annual Curated Data Release 2.01, https://data-archive.nimh.nih.gov/abcd), recruited 11,875 children between 9 and 10 years of age from 21 sites across the United States[37]. The study conforms to each site's Institutional Review Board's rules and procedures. All participants provide informed consent (parents) or informed assent (children). More details of the subjects and the data collection are provided at the ABCD website (https://abcdstudy.org/scientists/protocols)[37]. MRI data in the ABCD study were collected from different 3 T scanner platforms (i.e., Siemens Prisma, General Electric (GE) MR750 and Philips Achieva dStream). High resolution 3-dimensional T1-weighted images (1.0 mm isotropic voxel size) were acquired with harmonised imaging protocols (https://abcdstudy.org/images/Protocol_Imaging_Sequences.pdf). The processing of VBM was the same as that in the IMAGEN project. The rule-breaking score and conduct problem score were rated according to the child behaviour checklist (CBCL, parent-rated)[67].

To ensure homogeneity of the datasets, only self-reported white people were included to validate the relationship between the GMV of left vmPFC and rule-breaking behaviour and discover the genetic factors underlying the neurobehavioural circuit (see Fig. S8 for the plot of the first two PCs projected to 1000 Genome data). Therefore, 4417 participants with the quality-controlled brain images at baseline year and complete behavioural data at both baseline year and one-year follow-up interview were included. Similarly, sex, handedness, site, BMI, TIV, parents' education, and family income were regressed out as covariates in the relevant analyses.

## GWAS for the GMV of left vmPFC

To discover the genetic factors underlying the neurobehavioural circuit from the left vmPFC to rule-breaking behaviour, we performed a meta-analysis of GWAS for GMV of the left vmPFC. For ABCD cohorts, we performed the same quality control and imputation procedures as the above IMAGEN study. After imputation, 4244228 SNPs of 4390 participants (with complete phenotypes and with no siblings) from ABCD were available. GWAS was performed using PLINK with sex, research sites and top 20 ancestry principal components of ABCD cohorts as covariates (see Fig. S8 for the plot of the first two PCs projected to 1000 Genome data). The meta-analysis was performed using METAL software (https://csg.sph.umich.edu/abecasis/metal), and the genome-wide significant threshold was set as 5E-8.

## Human Connectome Project

Total 330 regular smokers (ages 22-37 years, mean age 29.49 y, 152 females) from the Human Connectome Project[40] were included in our current study. The HCP consortium is a public shared large-scale neuroimaging dataset and the details on the inclusion and exclusion criteria of HCP consortium were provided in the previous study[40] and the HCP website (https://www.humanconnectome.org). High resolution 3-dimensional T1-weighted images (0.8 mm isotropic voxel size) were scanned on a 3 T Siemens connectome-Skyra scanner. The processing of VBM was the same as that in the IMAGEN project. Smoking behaviour and the level of nicotine dependence were assessed according to the Fagerström Test for Nicotine Dependence (FTND) and DSM.

## Reporting summary

Further information on research design is available in the Nature Portfolio Reporting Summary linked to this article.

## Data availability

The IMAGEN project data are available from a dedicated database: https://imagen-project.org. ABCD data are available from a dedicated database: https://abcdstudy.org. HCP data are available from a dedicated database: https://www.humanconnectome.org. All data needed to evaluate the conclusions in this study are present in the paper and/or the Supplementary Information. Source data are provided with this paper (https://doi.org/10.6084/m9.figshare.23508369)[68].

 

## Code availability

Custom code that supports the findings of this study is available here (https://github.com/Shitong-Xiang/Smoking.git). Additional information related to this paper may be requested from the authors.

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

## Acknowledgements

We thank Professor David Nutt and Professor Stephan Ripke for their helpful comments on the manuscript. We thank the IMAGEN Consortium for providing the discover data. This work received support from the following sources: National Key R and D Program of China (2021YFC2501402 [to T.J.], 2022CSJGG1000 [to T.J.], 2019YFA0709501 [to T.J.], 2019YFA0709502 [to J.F.], 2018YFC1312900 [to T.J.] and 2018YFC1312904 [to J.F.]), the National Natural Science Foundation of China (T2122005 [to T.J.], 82150710554 [to G.S.] and 81801773 [to T.J.]), the Shanghai Pujiang Project (18PJ1400900 [to T.J.]), Guangdong Key Research and Development Project (2018B030335001 [to J.F.]), the European Union-funded FP6 Integrated Project IMAGEN (Reinforcement-related behaviour in normal brain function and psychopathology) (LSHM-CT- 2007-037286 [to G.S.]), the 111 Project (B18015 [to J.F.]), the key project of Shanghai Science and Technology (16JC1420402 [to J.F.]), Shanghai Municipal Science and Technology Major Project (2018SHZDZX01 [to J.F.]), Zhangjiang Lab [to J.F.], Shanghai Center for Brain Science and Brain-Inspired Technology [to J.F.], the European Research Area Network on Illicit Drugs (ERANID) (Understanding the Interplay between Cultural, Biological and Subjective Factors in Drug Use Pathways, PR-ST-0416-10004 [to G.S.]), Human Brain Project (HBP SGA 2, 785907, and HBP SGA 3, 945539 [to G.S.]), the Medical Research Council Grant 'c-VEDA' (Consortium on Vulnerability to Externalizing Disorders and Addictions, MR/N000390/1 [to G.S.]), the National Institute of Health (NIH) (A decentralized macro and micro gene-by-environment interaction analysis of substance use behavior and its brain biomarkers, R01DA049238 [to G.S.]), the National Institute for Health Research (NIHR) Biomedical Research Centre at South London and Maudsley NHS Foundation Trust and King's College London, the Bundesministeriumfür Bildung und Forschung (BMBF grants 01GS08152; 01EV0711 [to G.S.]; the European Union and UKRI funded project 'environMENTAL' (grants 101057429 [to G.S.] and 10038599 [to S.D.]); Forschungsnetz AERIAL 01EE1406A, 01EE1406B and Forschungsnetz IMAC-Mind 01GL1745B [to G.S.]), the Medical Research Foundation and Medical Research Council (grants MR/R00465X/1, MR/S020306/1 and MRF-058-0009-RG-DESR-C0759 [to S.D.]), the National Institutes of Health (NIH) funded ENIGMA (grants 5U54EB020403-05 and 1R56AG058854-01 [to S.D.]). Further support was provided by grants from: - the ANR (ANR-12-SAMA-0004 [to M.-L.P.M.], AAPG2019 – GeBra [to J.-L.M.]), the Eranet Neuron (AF12-NEURO0008-01 - WM2NA; and ANR-18-NEURO00002-01 – ADORe [to J.-L.M.]), the Fondation de France (00081242 [to J.-L.M.]), the Fondation pour la Recherche Médicale (DPA20140629802 [to J.-L.M.]), the Mission Interministérielle de Lutte-contre-les-Drogues-et-les-Conduites-Addictives (MILDECA [to J.-L.M.]), the Assistance-Publique-Hôpitaux-de-Paris and INSERM (interface grant [to M.-L.P.M.]), Paris Sud University IDEX 2012 [to J.-L.M.], the Fondation de l'Avenir (grant AP-RM-17-013 [to M.-L.P.M.]), the Fédération pour la Recherche sur le Cerveau; the National Institutes of Health, Science Foundation Ireland (16/ERCD/3797 [to R.W.]). The views expressed in this article are those of the authors and not necessarily those of the national funding agencies or ERANID.

## Author contributions

T.J., J.F., and T.W.R. conceptualized the study; S.X. and T.J. designed the analytic approach; S.X. analysed the data; S.X. and T.J. wrote the

manuscript; C.X., and W.C. helped in preprocessing the neuroimaging and genetic data; C.X. helped visualization; B.C., H.G. and B.J.S. helped in interpreting the results; T.W.R., B.J.S. and J.F. revised the first draft; T.B., G.J.B., A.L.W.B., C.B., S.D., H.F., A.G., P.A.G., R.B., J.-L.M., M.-L.P.M., F.N., D.P.O., L.P., S.H, J.H.F., M.N.S., N.V., H.W., R.W., H.G., and G.S. were the principal investigators of IMAGEN Consortium; Imaging, genetic and behavioural data in the IMAGEN project were acquired and provided by the IMAGEN Consortium; All authors critically revised the manuscript. G.S., B.J.S., T.W.R., and J.F. contributed equally to this paper.

## Competing interests

Tobias Banaschewski served in an advisory or consultancy role for eye level, Infectopharm, Lundbeck, Medice, Neurim Pharmaceuticals, Oberberg GmbH, Roche, and Takeda. He received conference support or speaker's fee by Janssen, Medice and Takeda. He received royalties from Hogrefe, Kohlhammer, CIP Medien, Oxford University Press; the present work is unrelated to these relationships. Dr. Barker has received honoraria from General Electric Healthcare for teaching on scanner programming courses. Dr. Poustka served in an advisory or consultancy role for Roche and Viforpharm and received speaker's fee by Shire. She received royalties from Hogrefe, Kohlhammer and Schattauer. The present work is unrelated to the above grants and relationships.The other authors report no biomedical financial interests or potential non-financial competing interests.

## Ethical approval

The IMAGEN project was approved by local ethics research committees at each research site: King's College London, University of Nottingham, Trinity College Dublin, University of Heidelberg, Technische Universität Dresden, Commissariatà l'Energie Atomique et aux Energies Alternatives and University Medical Center. Informed consent was sought from all participants and a parent/guardian of each participant. The ABCD study conforms to each site's institutional review board's rules and procedures, and all participants provided informed consent (parents) or informed assent (children). The HCP Consortium obtained full informed consent from all participants, and research procedures and ethical guidelines were followed in accordance with the institutional review boards. The UK Biobank has research tissue bank approval from the North West Multi-centre Research Ethics Committee and provided oversight for this study. Written informed consent was obtained from all participants.

## Additional information

[1]Institute for Science and Technology of Brain-inspired Intelligence (ISTBI), Fudan University, Shanghai, China. [2]Key Laboratory of Computational Neuroscience and Brain-Inspired Intelligence (Fudan University), Ministry of Education, Shanghai, China. [3]Centre for Population Neuroscience and Stratified Medicine (PONS Centre), ISTBI, Fudan University, Shanghai, China. [4]Social Genetic and Developmental Psychiatry Centre, Institute of Psychiatry, Psychology and Neuroscience, King's College London, London, UK. [5]Departments of Psychiatry and Psychology, University of Vermont, Burlington, VT, USA. [6]Department of Child and Adolescent Psychiatry and Psychotherapy, Central Institute of Mental Health, Medical Faculty Mannheim, Heidelberg University, Mannheim, Germany. [7]Department of Neuroimaging, Institute of Psychiatry, Psychology and Neuroscience, King's College London, London, UK. [8]Discipline of Psychiatry, School of Medicine and Trinity College Institute of Neuroscience, Trinity College Dublin, Dublin, Ireland. [9]University Medical Centre Hamburg-Eppendorf, Hamburg, Germany. [10]Institute of Cognitive and Clinical Neuroscience, Central Institute of Mental Health, Medical Faculty Mannheim, Heidelberg University, Mannheim, Germany. [11]Department of Psychology, School of Social Sciences, University of Mannheim, Mannheim, Germany. [12]NeuroSpin, C.E.A., Université Paris-Saclay, Gif-sur-Yvette, France. [13]Sir Peter Mansfield Imaging Centre School of Physics and Astronomy, University of Nottingham, Nottingham, UK. [14]Physikalisch-Technische Bundesanstalt (PTB), Berlin, Germany. [15]Institut National de la Santé et de la Recherche Médicale, INSERM U1299 'Trajectoires développementales en psychiatrie', Université Paris-Saclay, Ecole Normale supérieure Paris-Saclay, CNRS UMR9010, Centre Borelli, Gif-sur-Yvette, France. [16]AP-HP, Sorbonne Université, Department of Child and Adolescent Psychiatry, Pitié-Salpêtrière Hospital, Paris, France. [17]Institute of Medical Psychology and Medical Sociology, University Medical Center Schleswig-Holstein, Kiel University, Kiel, Germany. [18]Department of Child and Adolescent Psychiatry, Center for Psychosocial Medicine, University Hospital Heidelberg, Heidelberg, Germany. [19]Department of Psychiatry and Psychotherapy, Technische Universität Dresden, Dresden, Germany. [20]Department of Psychiatry and Neurosciences, Charité–Universitätsmedizin Berlin, corporate member of Freie Universität BerlinHumboldt-Universität zu Berlin, and Berlin Institute of Health, Berlin, Germany. [21]School of Psychology and Global Brain Health Institute, Trinity College Dublin, Dublin, Ireland. [22]Centre for Population Neuroscience and Stratified Medicine (PONS Centre), Charité University Medicine Berlin, Berlin, Germany. [23]Department of Psychiatry and Behavioural and Clinical Neuroscience Institute, University of Cambridge, Cambridge, UK. [24]Department of Psychology and Behavioural and Clinical Neuroscience Institute, University of Cambridge, Cambridge, UK. [25]Department of Computer Science, University of Warwick, Coventry, United Kingdom. [26]Zhangjiang Fudan International Innovation Center, Shanghai, China. [27]These authors contributed equally: Shitong Xiang, Tianye Jia. ✉e-mail: tianyejia@fudan.edu.cn; twr2@cam.ac.uk; jianfeng64@gmail.com

## IMAGEN Consortium

Tianye Jia [1,2,3,4,27] ✉, Tobias Banaschewski [6], Gareth J. Barker [7], Arun L. W. Bokde [8], Christian Büchel [9], Sylvane Desrivières [4], Herta Flor [10,11], Antoine Grigis [12], Penny A. Gowland [13], Rüdiger Brühl [14], Jean-Luc Martinot [15], Marie-Laure Paillère Martinot [15,16], Frauke Nees [6,10,17], Dimitri Papadopoulos Orfanos [11], Luise Poustka [18], Sarah Hohmann [6], Juliane H. Fröhner [19], Michael N. Smolka [19], Nilakshi Vaidya [20], Henrik Walter [20], Robert Whelan [21], Hugh Garavan [5] & Gunter Schumann [1,3,20,22]

