## [Peer Review File · Nature Communications]

Association between vmPFC gray matter volume and smoking initiation in adolescentsReviewer #1 (Remarks to the Author):

This is a well written manuscript elucidating the causal relationship between smoking during early adolescence and neurodevelopment. The topic is of importance to the field and the findings are novel and worth to be published.

However, enthusiasm is hampered by several methodological aspects of the study that are not clear in the paper. Particularly those relating to the assessment of the smoking behaviour/characteristics of participants. The literature overview is also incomplete.

The authors should therefore revise the manuscript based on my comments and answer my queries listed below.

(1) The literature overview in the introduction section is incomplete and animal models investigating the effect of nicotine on the adolescent brain are not mentioned. The authors should cite the following relevant studies.

References:

Mashhoon, Y., Betts, J., Farmer, S. L., & Lukas, S. E. (2018). Early onset tobacco cigarette smokers exhibit deficits in response inhibition and sustained attention. *Drug and Alcohol Dependence*, 184, 48-56.

Conti, A. A., & Baldacchino, A. M. (2021). Neuroanatomical Correlates of Impulsive Choices and Risky Decision Making in Young Chronic Tobacco Smokers: A Voxel-Based Morphometry Study. *Frontiers in psychiatry*, 12.

Yuan, M., Cross, S. J., Loughlin, S. E., & Leslie, F. M. (2015). Nicotine and the adolescent brain. *The Journal of physiology*, 593(16), 3397-3412.

Goriounova, N. A., & Mansvelder, H. D. (2012). Short-and long-term consequences of nicotine exposure during adolescence for prefrontal cortex neuronal network function. *Cold Spring Harbor perspectives in medicine*, 2(12), a012120.

Hu, T., Gall, S. L., Widome, R., Bazzano, L. A., Burns, T. L., Daniels, S. R., ... & Jacobs Jr, D. R. (2020). Childhood/adolescent smoking and adult smoking and cessation: the international childhood cardiovascular cohort (i3C) consortium. *Journal of the American Heart Association*, 9(7), e014381.

Kendler, K. S., Myers, J., Damaj, M. I., & Chen, X. (2013). Early smoking onset and risk for subsequent nicotine dependence: a monozygotic co-twin control study. *American Journal of Psychiatry*, 170(4), 408-413.

Paul, S. L., Blizzard, L., Patton, G. C., Dwyer, T., & Venn, A. (2008). Parental smoking and smoking experimentation in childhood increase the risk of being a smoker 20 years later: the Childhood Determinants of Adult Health Study. *Addiction*, 103(5), 846-853.

Leslie, F. M. (2020). Unique, long-term effects of nicotine on adolescent brain. *Pharmacology Biochemistry and Behavior*, 197, 173010.

Goldstein, R. Z., & Volkow, N. D. (2011). Dysfunction of the prefrontal cortex in addiction: neuroimaging findings and clinical implications. *Nature reviews neuroscience*, 12(11), 652-669.

(2)

Line 27 page 3.

“On the other hand, addiction, including nicotine dependence, may cause damage to the brain and accelerate brain aging”

This statement is vague and not very informative. The authors should definitively be more precise here. Is it nicotine dependence that causes damage to the brain or the neurotoxic properties of nicotine (see previous animal models) and of other chemical compounds present in tobacco cigarettes and/or resulting from tobacco combustion? (e.g. cadmium, benzene, lead...).

(3)

Considering the strong relationship between tobacco smoking during adolescence and comorbid psychiatric conditions (e.g. ADHD), it would be useful to know which screening tools (e.g. K-SADS) were employed to assess for/exclude these disorders at all study time points (14, 19, 23).

(4)

Line 310 page 13

“Cigarette smoking for each time point (i.e. of age 14, 19 or 23) was measured by the item "How many occasions during your lifetime have you smoked cigarettes?" from the Fagerstrom Test for Nicotine Dependence (FTND) 43 312 . Participants with scores greater than 0 were considered smokers at the interview. For instance, participants with smoking experience at the baseline interview (14 years old) were assigned to the baseline smokers group (BL-S).”

Are the authors investigating the causal relationship between neurodevelopment and chronic smoking, or between neurodevelopment and occasional smoking or both? Were participants considered smokers and assigned to the baseline smoker group even though they smoked, for example, only 1 tobacco cigarette 2 years before the interview?

In the paper, the authors should justify their choice for utilising just one item from the FTND (an item that assesses lifetime smoking experience and not recent smoking exposure nor frequency of smoking) to categorize participants as smokers.

Also, this item does not seem to be part of the FTND

https://cde.drugabuse.gov/sites/nida_cde/files/FagerstromTest_2014Mar24.pdf.

(5)

Line 50 page 4

“After quality control, this study included 807 (36.4%, 444 female participants [55%]) participants with complete structural images and 52 behavioural scores at both BL and FU. Among them, 181 participants, i.e., the baseline 53 smokers (BL-S), smoked before the baseline interview.”

How did the authors know that participants smoked before the baseline interview? Were objective methods for smoking detection (e.g. CO breath tests, saliva cotinine test) employed during the baseline visit or they relied solely on self-report measures?

Again, it is not clear what are the smoking characteristics of participants at all time points (mean number of cigarettes smoked per day, pack years, level of nicotine dependence..).

The Baseline smoking characteristics of participants should be included in Table 1.

(6)

Discussion section

It would be useful to discuss the findings of this study also in the context of the vaping epidemic among adolescents, which is a current public health issue. I would insert as a future direction the need to conduct similar studies investigating the causal relationship between e-cig use and brain development.

Reviewer #2 (Remarks to the Author):

Xiang and co-authors provide an elegant analysis of the IMAGEN dataset, including partial replication/extension in ABCD. Specifically, they look at grey matter structural predictors of smoking initiation and continued use in IMAGEN, and examine associations with risk behaviors/impulsivity, as measured by the SURPS, as well as genetic associations. The brain-genetic associations (PRS scores) and brain-impulsivity findings were tested in ABCD were also significant in this external sample but the effect sizes were very small ($r < -.02$), raising concerns about actual generalizability. Other methodological concerns include multiple comparison testing, particularly for all of the different individual SURPS items considered and the simple definition of smoking (yes/no at each time point). Given well-documented sex differences in neurodevelopment, I was also surprised that there was no attempt to look at sex interactions. Some of the methods are also unclear, for example, p.18, line 430 'only white men were included' versus p.18 line 435 in which sex is listed as a covariate. The Discussion lacks acknowledgement of clear limitations, including absence of sex comparisons and the overall relatively small effect sizes for most analyses.

Reviewer #3 (Remarks to the Author):

The current manuscript from Xiang et al. describes an analysis of smoking behaviors and gray matter volume in the longitudinal IMAGEN sample (assessed at ages 14, 19, and 23, though most analyses reported are comparisons across baseline (age 14) and the first follow-up (age 19)). The authors were interested in testing whether there are structural brain differences that predispose one to initiate smoking, and/or whether sustained smoking behavior results in structural brain differences. They find evidence that reduced gray matter volume (GMV) in the left vmPMC is potentially causal for smoking initiation (via rule-breaking). Conversely, they find that sustained smoking behavior may be causal for reduction in GMV in the right vmPFC.

This article addresses important questions about the relationships between smoking and brain development, leveraging a longitudinal sample with genetic, neuroimaging, and smoking (and other behavioral) measures to address causality. This is a real strength of the study, and I think the results from this study will be of great interest to the field. However, I have several suggestions for improvement, detailed below (Note: I am not an expert in neuroimaging approaches, so I have not commented as much on those aspects of the paper, but have mainly focused on the genetic analyses.)

Comments:

- The genetic analyses include GWAS of GMV, rule-breaking, and smoking quantity in subjects of the IMAGEN study who were not included in the other analyses due to lack of

neuroimaging data at follow-up (N = 1,026). "Valid PRS" were then scored in the analytic sample of IMAGEN (N = 752) using these GWAS results, and used as indicators in a Mendelian Randomization approach. Could the authors please provide more analysis details about these GWAS? For example, did the authors generally follow the same quality control procedures as they did for the larger GWAS meta-analysis of GMV of the left vmPFC? What covariates were included in the GWAS? Did the authors ensure that none of the individuals in the "discovery" GWAS (N = 1,026) were genetically related to those in the "target" PRS sample (N = 752)?

- The authors also conducted a separate GWAS meta-analysis of GMV of the left vmPFC in the full IMAGEN cohort (N = 1,778) and the ABCD cohort (N = 4,390). However, the top SNP in their analysis did not replicate in the UK Biobank cohort (N = 38,677), and neither was the PRS associated in the UK Biobank. The authors speculate that this is evidence for time-specific genetic effects on left vmPFC GMV, and while that certainly might be true, couldn't this also potentially represent a false positive in the authors' GWAS? I believe neuroimaging phenotypes show slightly larger genetic effect sizes on average, compared to behavioral traits, but the GWAS N (meta-analyzed = 6,168) still seems relatively low-powered for this analysis. It would be helpful if the authors could provide a sensitivity analysis that estimates how much statistical power they had to detect reasonable effect sizes based on previous literature. Relatedly, it might be helpful for the authors to present the effect sizes of the top SNP(s) in the IMAGEN and ABCD cohorts, either in a forest plot or a supplementary table, so readers can assess whether the signal is being driven primarily by one cohort or by both. Given the relatively low statistical power of their samples, I would opt not to include in the text the associations of the 'G' allele at the top SNP with rule-breaking, conduct problems, or smoking behavior, especially as these p-values don't seem to be corrected for any multiple testing and range from 0.02 to 0.049.

- The authors' creation of "valid PRS" seems appropriate, although it would be helpful if more information was provided on this method – has this exact approach been used before? (The authors reference a few MR papers, including Burgess et al. (2020), but it does not appear that the Burgess et al. paper used this PRS approach.) If the authors are introducing a new MR method, it might increase readers' confidence in this new approach if the authors were to demonstrate its feasibility first with well-established causal relationships (e.g., LDL cholesterol and heart disease), before applying their method to the traits of interest (GMV of vmPFC, smoking behaviors, sensation-seeking).

- In the Methods section, the authors describe the ABCD Study analytic sample (for the GWAS meta-analysis) as being confined to "white men" and say that they focused on "white people from both IMAGEN and ABCD cohorts". First, given the age of the sample, it might be more accurate to describe the ABCD sample as consisting of white males (instead of men), or male children whose parents reported their child's race as being "white". Second, I assume the authors are using self-reported race here, but it is more typical for genetic analyses to use genetic principal components to define more homogeneous genetic ancestry groups without the use of self-reported race, or to "confirm" self-reported race using genetic principal components (i.e., cross-check self-reported race with clustering on genetic PCs). Could the authors comment on their choice of approach here? It might also be helpful to see a PCA plot of their analytic samples, with the PCs projected onto the major 1000 Genomes ancestry groups. Furthermore, the authors don't appear to have corrected for genetic PCs in the IMAGEN study – yet even in a "homogeneous" population, correcting for PCs is important to control for genetic structure (<https://doi.org/10.1038/nature07331>).

- It was unclear to me whether the authors tested associations only with gray matter volume, or whether they tested other modalities as well. If they did not test other modalities, I think it would be helpful for the authors to provide a bit more explanation of their study motivations/hypotheses in the Introduction and justify their choice to focus only on gray matter volume. Similarly, it was unclear to what extent the authors tested multiple regions and only saw significant effects in the left vmPFC and right vmPFC, and thus only reported on those effects, or whether they chose *a priori* to focus on these regions. This should be more clearly spelled out in the manuscript.

Minor comments:

- The mediation model shown in Extended Data Figure 2 is a bit confusing to me – it seems to show follow-up smoking behavior being the mediator of a relationship between left vmPFC at baseline and sensation seeking at follow-up, but this seems counterintuitive – wouldn't we expect sensation seeking to be the mediator of a relationship between left vmPFC and smoking?
- The abbreviation "MNI" is introduced in the Results section with no explanation (e.g., "Peak MNI: [-2, 44, 6]") – could the authors provide the full version of this abbreviation at first mention?
- Similarly, Table 1 could use explanations for all of the abbreviations (e.g., TIV, BMI, WISCIV).
- For several of the Extended Data Tables, the p-values are given as "0.000". Could the authors please present a p-value in scientific notation here?
- Inconsistent scientific notation: at one place in the text, a p-value is given as "1E-3", whereas other times similar p-values are given as < 0.001 .

REVIEWER COMMENTS

Reviewer #1 (Remarks to the Author):

This is a well written manuscript elucidating the causal relationship between smoking during early adolescence and neurodevelopment. The topic is of importance to the field and the findings are novel and worth to be published.

However, enthusiasm is hampered by several methodological aspects of the study that are not clear in the paper. Particularly those relating to the assessment of the smoking behaviour/characteristics of participants. The literature overview is also incomplete.

The authors should therefore revise the manuscript based on my comments and answer my queries listed below.

(1) The literature overview in the introduction section is incomplete and animal models investigating the effect of nicotine on the adolescent brain are not mentioned.

The authors should cite the following relevant studies.

References:

Mashhoon, Y., Betts, J., Farmer, S. L., & Lukas, S. E. (2018). Early onset tobacco cigarette smokers exhibit deficits in response inhibition and sustained attention. *Drug and Alcohol Dependence*, 184, 48-56.

Conti, A. A., & Baldacchino, A. M. (2021). Neuroanatomical Correlates of Impulsive Choices and Risky Decision Making in Young Chronic Tobacco Smokers: A Voxel-Based Morphometry Study. *Frontiers in psychiatry*, 12.

Yuan, M., Cross, S. J., Loughlin, S. E., & Leslie, F. M. (2015). Nicotine and the adolescent brain. *The Journal of physiology*, 593(16), 3397-3412.

Goriounova, N. A., & Mansvelder, H. D. (2012). Short-and long-term consequences of nicotine exposure during adolescence for prefrontal cortex neuronal network function. *Cold Spring Harbor perspectives in medicine*, 2(12), a012120.

Hu, T., Gall, S. L., Widome, R., Bazzano, L. A., Burns, T. L., Daniels, S. R., ... & Jacobs Jr, D. R. (2020). Childhood/adolescent smoking and adult smoking and cessation: the international childhood cardiovascular cohort (i3C) consortium. *Journal of the American Heart Association*, 9(7), e014381. Kendler, K. S., Myers, J., Damaj, M. I., & Chen, X. (2013). Early smoking onset and risk for subsequent nicotine dependence: a monozygotic co-twin control study. *American Journal of Psychiatry*, 170(4), 408-413.

Paul, S. L., Blizzard, L., Patton, G. C., Dwyer, T., & Venn, A. (2008). Parental smoking and smoking experimentation in childhood increase the risk of being a smoker 20 years later: the Childhood

Determinants of Adult Health Study. *Addiction*, 103, 846-853.

Leslie, F. M. (2020). Unique, long-term effects of nicotine on adolescent brain. *Pharmacology Biochemistry and Behavior*, 197, 173010.

Goldstein, R. Z., & Volkow, N. D. (2011). Dysfunction of the prefrontal cortex in addiction: neuroimaging findings and clinical implications. *Nature reviews neuroscience*, 12(11), 652-669.

Reply: We thank the reviewer for this very useful suggestion and have now included all these references in Introduction:

"... Smoking initiation is most likely to occur during adolescence, and previous studies in human cohort and animal models have suggested that early nicotine exposure during adolescence could directly increase the risk of nicotine dependence in the future ⁸⁻¹¹... There is a long latency from substance use to disorder (i.e., addiction), which offers a significant window of opportunity for clinical interventions. However, treatment efforts have focused almost exclusively on those with serious, usually chronic addictions, virtually ignoring the much larger population with pre-addiction ¹⁶. Therefore, this tremendous burden on public health calls for further understanding of the biological mechanisms contributing to smoking initiation and early-stage sustenance...Its disrupted development has been implicated as a trigger for maladaptive behaviour, such as addiction ^{19,20,23}. On the other hand, substance use, including nicotine exposure, may cause damage to the brain and accelerate brain aging potentially through its neurotoxic properties, indirectly exacerbated by excessive smoking ^{8,10}. Such neurotoxic effects may also affect the reinforcement system itself and induce other forms of substance dependence ^{24,25}. Further, there are significant age differences in many of the acute neurobehavioural impact of nicotine ²⁶. For instance, early onset smokers exhibit deficits in reward processing and response inhibition, whereas other behavioural effects, such as physical craving for nicotine following withdrawal are greater in adults ²⁶⁻²⁹. While several associations of cigarette smoking with cognitive functions and brain structures have been established ³⁰⁻³², no consensus has been reached on the exact causal relationship between brain development and smoking in adolescence and its underlying neurobehavioural mechanisms remain elusive."

(2) Line 27 page 3.

"On the other hand, addiction, including nicotine dependence, may cause damage to the brain and accelerate brain aging"

This statement is vague and not very informative. The authors should definitively be more precise here. Is it nicotine dependence that causes damage to the brain or the neurotoxic properties of nicotine (see previous animal models) and of other chemical compounds present in tobacco cigarettes and/or resulting from tobacco combustion? (e.g. cadmium, benzene, lead...).

Reply: We have revised this statement as follows:

"On the other hand, substance use, including nicotine exposure, may cause damage to the brain and accelerate brain aging potentially through its neurotoxic properties, indirectly exacerbated by excessive smoking ^{8,10}. Such neurotoxic effects may also affect the reinforcement system itself and induce other forms of substance dependence ^{24,25}."

(3) Considering the strong relationship between tobacco smoking during adolescence and comorbid psychiatric conditions (e.g. ADHD), it would be useful to know which screening tools (e.g. K-SADS) were employed to assess for/exclude these disorders at all study time points (14, 19, 23).

Reply: At the baseline interview, mental health was assessed with Development and Well-being Assessment (DAWBA, parent-rated) in the IMAGEN project. According to the Diagnostic and Statistical Manual of Mental Disorders (DSM), the computer prediction scores for common mental disorders (including Attention Deficit and Hyperactivity Disorder, Conduct Disorder, Oppositional Defiant Disorder, Generalised Anxiety and Major Depression) were rated, and participants with diagnostic risk greater than 50% were reported as cases. As follows, there were no significant differences in mental health between the smokers and controls.

Characteristics	All Sample (n = 807)	Con Group (n = 260)	BL-S Group (n = 181)	FU-S Group (n = 366)
Mental Disorders				
Attention Deficit and Hyperactivity Disorder	37 (4.58%)	12 (4.62%)	8 (4.42%)	17 (4.64%)
Conduct Disorder	28 (3.47%)	8 (3.08%)	7 (3.87%)	14 (3.83%)
Oppositional Defiant Disorder	15 (1.86%)	4 (1.54%)	4 (2.21%)	7 (1.91%)
Generalised Anxiety	14 (1.73%)	5 (1.92%)	3 (1.66%)	6 (1.64%)
Major Depression	26 (3.22%)	8 (3.08%)	6 (3.31%)	12 (3.28%)

The above information has been included in the revised Table 1.

(4) Line 310 page 13

"Cigarette smoking for each time point (i.e. of age 14, 19 or 23) was measured by the item "How many occasions during your lifetime have you smoked cigarettes?" from the Fagerstrom Test for Nicotine Dependence (FTND) 43 312 . Participants with scores greater than 0 were considered smokers at the interview. For instance, participants with smoking experience at the baseline interview (14 years old) were assigned to the baseline smokers group (BL-S)."

Are the authors investigating the causal relationship between neurodevelopment and chronic smoking, or between neurodevelopment and occasional smoking or both? Were participants considered smokers and assigned to the baseline smoker group even though they smoked, for example, only 1 tobacco cigarette 2 years before the interview?

In the paper, the authors should justify their choice for utilising just one item from the FTND (an item that assesses lifetime smoking experience and not recent smoking exposure nor frequency of smoking) to categorize participants as smokers.

Also, this item does not seem to be part of the FTND https://cde.drugabuse.gov/sites/nida_cde/files/FagerstromTest_2014Mar24.pdf.

Reply: We thank the reviewers for these constructive suggestions. Please find our detailed replies as follows:

(1) In the present study, utilising the longitudinal IMAGEN data (i.e., from adolescence to early adulthood), we aimed to understand the neurobehavioural mechanisms contributing to both the **initiation** (with even an occasional use) and **sustenance** (i.e., reinforcement processes leading eventually to habitual chronic use) of smoking at this critical early period. Particularly, we would like to highlight that both smoking **initiation** (for instance, the early exposure) and **sustenance** during adolescence are crucial for later nicotine dependence, and we have extended the relevant sentences in Introduction as follows:

"Smoking initiation is most likely to occur during adolescence, and previous studies in human cohorts and animal models have suggested that early nicotine exposure during adolescence could directly increase the risk of future nicotine dependence⁸⁻¹¹."

(2) Due to legally limited access to cigarettes or tobacco for European adolescents, participants in the IMAGEN study were unlikely to establish regular nicotine use behaviour (for instance, daily smoking) by age 19. Therefore, the longitudinal IMAGEN study (from ages 14 to 19) is most suitable to assess neuro-biomarkers underlying the early stage of nicotine use (for instance, the initiation and following maintenance).

Indeed, the majority of IMAGEN participants had never smoked (626/807) by the baseline interview (i.e., the baseline smokers reported at age 14), and 366 of the baseline non-smokers then reported smoking experience at the follow-up interview (i.e., the follow-up smokers only reported at age 19). Thus, we have a relatively balanced design if stratifying the IMAGEN participants based on their smoking experience (i.e., 260 non-smokers at both time points, 181 smokers before the baseline interview, and 366 smokers after the baseline interview). In contrast, only 11 (< 2%) and 58 (~7%) participants were reported as daily smokers at the baseline and follow-up, respectively.

(3) Nonetheless, we did conduct supplementary analyses to evaluate and confirm that the qualitatively identified neuro-biomarkers (i.e., from case-control comparisons: "baseline smokers vs controls" and "follow-up smokers vs controls") could also quantitatively influence lifetime occasions of smoking within both baseline and follow-up smokers. These results were shown in Fig. 1b, 2b & Table S2 from the original submission. For your reference, the relevant section of Table S2 is cited below:

Brain Features	Occasions of Smoking in Lifetime			
	BL-S Group ($n = 181$) at baseline		FU-S Group ($n = 366$) at follow-up	
	r	P one-tailed	r	P one-tailed
Left vmPFC volume at baseline	-0.170	0.016	-0.109	0.020
Right vmPFC Development	\	\	-0.126	0.010

To further alleviate the reviewer's concern regarding the occasional users, we now reconducted the quantitative analyses within both smoker groups but with 'occasional users' (with only 1 or 2 occasions) removed ($n = 87$ out of 181 at baseline and $n = 90$ out of 366 at follow-up), and the main results remained:

Brain Features	Occasions of Smoking in Lifetime (without Occasional Users)			
	BL-S Group ($n = 94$) at baseline		FU-S Group ($n = 276$) at follow-up	
	r	p _{one-tailed}	r	p _{one-tailed}
Left vmPFC volume at baseline	-0.208	0.037	-0.107	0.040
Right vmPFC Development	\	\	-0.135	0.014

Notably, the effect sizes were very similar with or without occasional users, hence indicating that the identified neuro-biomarkers are sensitive to the quantity of smoking experience, but not just the dichotomised with/without experience. We now included these new results in the manuscript:

“We further verified the above between-group results within groups BL_S and FU_S, and again found that the GMV of left vmPFC (of the 426 overlapped voxels) at baseline was not only associated with the smoking frequency at baseline in BL-S ($r_{166} = -0.17$, $p_{one-tailed} = 0.016$, Fig.1B upper) but could also predict the future smoking frequency in FU-S ($r_{351} = -0.11$, $p_{one-tailed} = 0.020$, Fig. 1b lower; also see Table S2). **Also, both above associations remained significant after excluding occasional users, i.e. with once or twice experience life-time ($r_{79} = -0.21$, $p_{one-tailed} = 0.037$ in BL-S and $r_{261} = -0.11$, $p_{one-tailed} = 0.040$ in FU-S).**”

“We then verified the between-group findings within FU-S and confirmed the association of higher smoking quantity with reduced GMV in the right vmPFC ($r_{349} = -0.13$, $p_{one-tailed} = 0.010$, Fig. 2b & Table S2). **Again, this association remained significant after removing occasional users ($r_{261} = -0.14$, $p_{one-tailed} = 0.014$).**”

It should also be noted that while about half of the baseline smokers (87/181) only smoked once or twice before attending the baseline interview, most baseline smokers (166/181) did report more smoking experience at the follow-up interview, thus indicating continuity in smoking behaviour (i.e., chronic use) among these early-exposure individuals.

Finally, while daily smokers were rare even at age 19 (as mentioned above), a majority of smokers (134/181 for BL-S and 280/366 for FU-S) did report smoking in the past 30 days before the follow-up interview (but not at the baseline interview, where only 31 participants reported). Thus, we conducted an additional analysis to see if identified neuro-biomarkers could predict the quantity/frequency of smoking in the last 30 days at the follow-up and again observed similar results:

Brain Features	Quantity of Smoking in the Last 30 Days at follow-up			
	BL-S Group ($n = 181$)		FU-S Group ($n = 366$)	
	r	p _{one-tailed}	r	p _{one-tailed}
Left vmPFC volume at baseline	-0.159	0.019	-0.091	0.040
Right vmPFC Development	\	\	-0.093	0.026

The above information is now included as follows:

*"The remaining 626 participants were further subdivided into a control group ($n = 260$) and a follow-up smoker group (FU-S, $n = 366$) based on whether they reported ever smoking at the FU interview. **Notably, while only 11 (< 2%) and 58 (~7%) participants were reported daily smokers at baseline and follow-up, respectively, a majority of smokers (134/181 for BL-S and 280/366 for FU-S) did report smoking in the past 30 days before the follow-up interview, indicating an ongoing progress towards more regular nicotine use.**"*

*"Also, both above associations remained after excluding occasional users, i.e. with once or twice experience life-time ($r_{79} = -0.21$, $p_{\text{one-tailed}} = 0.037$ in BL-S and $r_{261} = -0.11$, $p_{\text{one-tailed}} = 0.040$ in FU-S). **Finally, lower GMV in the left vmPFC could additionally predict higher future smoking quantities within 30 days before the follow-up interview ($r_{166} = -0.16$, $p_{\text{one-tailed}} = 0.019$ in BL-S and $r_{351} = -0.09$, $p_{\text{one-tailed}} = 0.040$ in FU-S).** The above results hence indicate reduced GMV in the left vmPFC as a highly sensitive risk factor for the initiation of future smoking behaviour."*

*"Again, this association remained significant after removing occasional users ($r_{259} = -0.14$, $p_{\text{one-tailed}} = 0.014$). **Additionally, reduced GMV in the right vmPFC was also associated with higher quantity of smoking in the last 30 days in FU-S ($r_{259} = -0.09$, $p_{\text{one-tailed}} = 0.026$).**"*

(4) The lifetime smoking frequency item was from the European School Survey Project on Alcohol and Other Drugs (ESPAD), but not the FTND. We thank the reviewer for pointing this out and have revised the relevant sentences accordingly. We are very sorry for the oversight here.

To elaborate on this matter further, in the IMAGEN study, both ESPAD and FTND were assessed using an integrated gateway system, where the FTND was only assessed if a participant was identified as a daily smoker based on the ESPAD result. We were confused by the explanatory document and mistakenly assumed all the smoking-related items were from the FTND. The relevant sentences in the Methods section have been revised as follows:

"Cigarette smoking for each time point (i.e., of age 14, 19 or 23) was measured by the item "How many occasions during your lifetime have you smoked cigarettes?" from the European School Survey Project on Alcohol and Other Drugs (ESPAD)."

(5) At last, in 330 daily smokers from the Human Connectome Project (Van Essen, Ugurbil et al. 2012), we found that smaller left vmPFC was associated (and stronger than the right vmPFC) with earlier initiation age of smoking. In contrast, smaller right vmPFC was associated (and stronger than the left vmPFC) with higher nicotine dependence (i.e., the FTND total score and DSM tobacco dependence-withdrawal). These new results (as summarised in Table S10 below) strongly support our main findings for the distinct roles of the left and right vmPFC in addictive behaviour as summarised in Discussion:

"Specifically, reduced GMV in the left vmPFC was associated with increased rule-breaking, which could well extend to the violation of social (e.g. parental, school) rules about smoking, possibly

because the loss of function in the left vmPFC leads to behavioural disinhibition due to the discounting of consequences of rule-breaking⁴¹. Following this initiation of smoking, the reduced GMV in the right vmPFC may subsequently sustain and thus strengthen smoking behaviour further by removing inhibitory constraints on reward seeking and heightening the hedonic experience of smoking.

Most notably, these new results have extended the role of the right vmPFC from early sustenance to dependence, especially the withdrawal effect. Such an extension was unsurprising, as both sustenance and dependence are related to the reinforcement process.

Table S10. The correlation between brain features and smoking dependence scores among regular smokers from the HCP dataset.

Tobacco Use and Dependence Score	L-vmPFC		R-vmPFC		Difference in r	
	r	$p_{one-tailed}$	r	$p_{one-tailed}$	Steiger's Z	P
FTND Total Score	-0.057	0.173	-0.135	0.012	2.243	0.025
Age first smoked a cigarette (even a puff)	0.153	0.005	0.084	0.061	1.989	0.047
DSM tobacco dependence - withdrawal	-0.008	0.448	-0.119	0.025	3.184	0.001

We have added the above results at the end of Results:

“Finally, while we cannot directly assess the impact of our neural biomarkers on addictive symptoms due to the minimal sample size of daily smokers in IMAGEN, we generalised our findings and verified the distinct role of both left and right vmPFC GMV in 330 regular/daily smokers from the Human Connectome Project⁴⁰. To elaborate, smaller left vmPFC was only associated ($r_{323} = 0.153$, $p_{one-tailed} = 0.005$) with earlier initiation age of smoking, stronger than that with the right vmPFC (Steiger’s $Z = 1.989$, $p_{one-tailed} = 0.047$). In contrast, smaller right vmPFC was associated with higher nicotine dependence (i.e., the FTND total score, $r_{323} = -0.135$, $p_{one-tailed} = 0.012$; and DSM tobacco withdrawal symptoms $r_{323} = -0.119$, $p_{one-tailed} = 0.025$), and both associations were stronger than the left vmPFC (Steiger’s $Z = 2.243$, $p_{one-tailed} = 0.025$, and Steiger’s $Z = 3.184$, $p_{one-tailed} = 0.001$, respectively) (Table S10). The above findings were highly consistent with the proposed initiation role of the left vmPFC and reinforcement role of the right vmPFC in this study.”

References:

Van Essen, D. C., Ugurbil, K., Auerbach, E., Barch, D., Behrens, T. E., Bucholz, R., . . . Consortium, W. U.-M. H. (2012). “The Human Connectome Project: a data acquisition perspective.” *Neuroimage*, 62(4), 2222-2231.

(5) Line 50 page 4

"After quality control, this study included 807 (36.4%, 444 female participants [55%]) participants with complete structural images and 52 behavioural scores at both BL and FU. Among them, 181 participants, i.e., the baseline 53 smokers (BL-S), smoked before the baseline interview."

How did the authors know that participants smoked before the baseline interview? Were objective

methods for smoking detection (e.g. CO breath tests, saliva cotinine test) employed during the baseline visit or they relied solely on self-report measures?

Again, it is not clear what are the smoking characteristics of participants at all time points (mean number of cigarettes smoked per day, pack years, level of nicotine dependence..).

The Baseline smoking characteristics of participants should be included in Table 1.

Reply: We determined whether participants smoked according to their self-report measures. The age of the first smoking, the occasion of smoking lifetime, and the quantity of smoking in the last 30 days were characterised. Due to the minimal sample size of daily smokers in the IMAGEN project (see replies to your question 4), the mean number of cigarettes smoked per day and the level of nicotine dependence were not available. The available smoking characteristics were summarised as follows and also included in the revised Table 1.

Characteristics	Con Group (n = 260)	BL-S Group (n = 181)	FU-S Group (n = 366)
Smoking Characteristics			
Age of First Smoking	\	12.97 (0.97)	16.08 (1.37)
Quantity of Smoking in Lifetime at Baseline	\	2.47 (1.78)	\
Quantity of Smoking in Lifetime at Follow-up	\	5.12 (1.52)	3.61 (2.01)
Quantity of Smoking in the Last 30 Days at Baseline	\	0.57 (1.08)	\
Quantity of Smoking in the Last 30 Days at Follow-up	\	2.32 (1.87)	1.20 (1.54)

(6) Discussion section

It would be useful to discuss the findings of this study also in the context of the vaping epidemic among adolescents, which is a current public health issue. I would insert as a future direction the need to conduct similar studies investigating the causal relationship between e-cig use and brain development.

Reply: Thanks for this helpful suggestion. We have now added the following sentence in the limitation section of Discussion:

"Furthermore, in the context of the vaping epidemic among adolescents, whether there are similar neurobehavioural mechanisms between neurodevelopment and e-cigarette use needs further investigation."

Reviewer #2 (Remarks to the Author):

Xiang and co-authors provide an elegant analysis of the IMAGEN dataset, including partial replication/extension in ABCD. Specifically, they look at grey matter structural predictors of smoking initiation and continued use in IMAGEN, and examine associations with risk behaviors/impulsivity, as measured by the SURPS, as well as genetic associations.

(1) The brain-genetic associations (PRS scores) and brain-impulsivity findings were tested in ABCD were also significant in this external sample but the effect sizes were very small ($r < -.02$), raising concerns about actual generalizability.

Reply:

First, we would like to clarify that both brain-genetic and brain-impulsive results have demonstrated good generalisability from IMAGEN to ABCD.

- 1) The PRS of GMV in the left vmPFC from the IMAGEN ($r_{max} = 0.399$, all $P < 0.001$) still has relatively large effect sizes when validated with GMV in the left vmPFC in the ABCD ($r_{max} = 0.162$, all $P < 0.001$, Table S7). Thus, the major brain-genetic finding demonstrated a very high generalisability.
- 2) The correlation between GMV in the left vmPFC and rule-breaking score was -0.14 at baseline in the IMAGEN ($P < 1.0E-4$, Table S4) and $-0.05 \sim -0.06$ in the ABCD ($P < 3.0E-4$, Fig. 3b), thus the brain-impulsivity findings also had a good generalisability.

However, we did observe reduced effect sizes when generalising both correlations from IMAGEN to ABCD. The reason behind could be multiple-folded: 1) the age gap between the ABCD (age 9-11) and IMAGEN (age 14-19), though not huge, could still have a negative impact on the generalisability; 2) different measurements were adopted for the rule-breaking score in the IMAGEN (from TCI questionnaire) and ABCD (from CBCL parent-rated); 3) the scan protocols and image qualities are different between the two studies as well. Most notably, the age gap effect could be readily observed in the longitudinal IMAGEN data (Table S5), where the PRS established at baseline (at age 14) has its prediction power (i.e., the correlation) reduced by more than half when predicting the same phenotypes measured five years later (from $r_{max} = 0.399$ at age 14 to $r_{max} = 0.149$ at age 19). For your further reference, we have now added a dedicated section to demonstrate that the genetic findings are indeed highly time sensitive (as part of our replies to Reviewer 3).

Second, the reviewer's concern was probably referring to the Mendelian randomisation results in the ABCD, where the findings' effect sizes ranged from $r = -0.024$ to -0.033 (Table S7). However, we would argue that these relatively smaller effect sizes (though still significant) were not a sign of doubtful generalisability, and instead, it matched our expectations. In the IMAGEN sample, the correlation between GMV in the left vmPFC and the rule-breaking score was $r = -0.14$ and the correlations between the corresponding PRS of GMV and the rule-breaking score (i.e., the causal inference using the Mendelian randomisation) was reduced by half to $r = -0.07$. In the ABCD sample,

however, the correlation between GMV in the left vmPFC and the rule-breaking score was found to be smaller (though still highly significant) with $r = -0.05 \sim -0.06$ (see above discussions for possible reasons). If following the same pattern in the IMAGEN data, we should also expect halved effect sizes (i.e., $-0.25 \sim -0.30$, which are still sufficient to be significant thanks to ABCD's large sample size) when correlating the corresponding GMV PRS to the rule-breaking score, exactly what we have observed (i.e., from $r = -0.024$ to -0.033 , Table S7).

We now explicitly provided all effect sizes and the corresponding p-values regarding generalisation to the ABCD cohort in the manuscript, and included the following clarification and limitation:

“Also, by applying the GWAS results obtained from IMAGEN, we verified the corresponding PRSs of both GMV in the left vmPFC ($r_{max} = 0.162$, $p_{one-tailed} < 0.001$) and rule-breaking behaviour ($r_{max} = 0.038$, $p_{one-tailed} = 0.006$) in the ABCD cohort (Table S7). The PRS findings hence indicate that similar genetic constructs exist for both cohorts. Further, MR analyses based on valid-PRS again reached a conclusion on the causality of GMV in the left vmPFC over rule-breaking behaviour ($r_{4365} < -0.025$, $p_{one-tailed} < 0.050$, Table S7) in the ABCD cohort, **where the effect sizes are roughly half of those in the imaging-behavioural associations ($r < -0.05$) similar to the observations from the IMAGEN cohort (i.e. from $r \sim -0.14$ to $r \sim -0.07$)**, thus suggesting this potential causal effect persisting throughout the whole adolescent developmental stage.”

“The other limitation of this study was that some of the genetic-behavioural findings in the generalisation cohort were relatively small, which could be largely due to the fact that the genetic constructs of left vmPFC GMV were highly time sensitive (Table S8 and Fig. S7).”

(2) Other methodological concerns include multiple comparison testing, particularly for all of the different individual SURPS items considered

Reply: We have now performed FDR correction for the p-values of the associations with the novelty seeking components and SURPS subitems, and our main results survive. The corresponding tables (Table S4 and Table S9) were also updated.

(3) and the simple definition of smoking (yes/no at each time point).

Reply: In our necessarily lengthy reply to address a similar question (Question 4) from Reviewer 1, we clarified that the IMAGEN data is primarily designed and most suitable to investigate the initiation and maintenance of early smoking behaviour, hence leading to the stratification based on smoking experience. In the reply, we also included a few old/new analyses to demonstrate that our main findings could be generalised to quantitative smoking measurements in the same dataset and smoking dependence in an independent HCP data. Please find more details there.

(4) Given well-documented sex differences in neurodevelopment, I was also surprised that there was no

attempt to look at sex interactions.

Reply: Following the reviewer's suggestion, we now re-investigated all the major findings within each sex and found no clear difference. Sex-stratified results were now added to the corresponding tables with significant findings with the full sample highlighted in **bold** (Table S1 and Table S2) as follows:

Summary statistics of gray matter volume and development in clusters (among males)							
Summary Statistics							
Clusters		Con Group (n = 106)		BL-S Group (n = 84)		FU-S Group (n = 173)	
		mean	std	mean	std	mean	std
Baseline	Left vmPFC	0.599	0.095	0.578	0.098	0.573	0.091
	Right vmPFC	0.487	0.091	0.469	0.090	0.479	0.082
Development	Left vmPFC	-3.09E-03	1.85E-03	-3.08E-03	1.44E-03	-3.35E-03	1.73E-03
	Right vmPFC	-2.50E-03	1.64E-03	-2.69E-03	1.31E-03	-3.54E-03	1.79E-03
Follow-up	Left vmPFC	0.577	0.077	0.526	0.094	0.544	0.076
	Right vmPFC	0.465	0.061	0.426	0.078	0.447	0.065
Group Comparison							
Clusters		BL-S vs Con Group			FU-S vs Con Group		
		T	p	Cohen's d	T	p	Cohen's d
Baseline	Left vmPFC	-1.913	0.029	-0.279	-3.145	0.001	-0.388
	Right vmPFC	-1.956	0.026	-0.286	-1.496	0.068	-0.185
Development	Left vmPFC	0.668	0.253	0.098	-0.478	0.316	-0.059
	Right vmPFC	-1.223	0.111	-0.179	-2.349	0.010	-0.290
Follow-up	Left vmPFC	-2.292	0.012	-0.335	-3.392	4.0E-4	-0.418
	Right vmPFC	-1.869	0.032	-0.273	-1.815	0.035	-0.224

Summary statistics of gray matter volume and development in clusters. (among females)							
Summary Statistics							
Clusters		Con Group (n = 154)		BL-S Group (n = 97)		FU-S Group (n = 193)	
		mean	std	mean	std	mean	std
Baseline	Left vmPFC	0.600	0.091	0.577	0.087	0.582	0.087
	Right vmPFC	0.485	0.072	0.476	0.073	0.475	0.082
Development	Left vmPFC	-4.84E-03	2.13E-03	-4.75E-03	2.04E-03	-4.95E-03	2.02E-03
	Right vmPFC	-3.00E-03	1.65E-03	-3.26E-03	1.61E-03	-3.92E-03	1.83E-03
Follow-up	Left vmPFC	0.584	0.082	0.552	0.091	0.564	0.081
	Right vmPFC	0.467	0.060	0.442	0.066	0.457	0.055
Group Comparison							
Clusters		BL-S vs Con Group			FU-S vs Con Group		
		T	p	Cohen's d	T	p	Cohen's d
Baseline	Left vmPFC	-1.701	0.045	-0.220	-2.034	0.021	-0.220
	Right vmPFC	-1.679	0.047	-0.218	-1.081	0.140	-0.117
Development	Left vmPFC	0.398	0.345	0.052	-0.845	0.199	-0.091
	Right vmPFC	-1.652	0.050	-0.214	-3.386	4.0E-4	-0.366
Follow-up	Left vmPFC	-1.792	0.037	-0.232	-1.974	0.025	-0.213
	Right vmPFC	-1.961	0.025	-0.254	-1.877	0.031	-0.203

Correlations between brain features and smoking (among males)				
Brain Features	Lifetime occasions of smoking at Baseline among BL-S Group (n = 84)		Lifetime occasions of smoking at Baseline among BL-S Group (n = 173)	
	r	p _{one-tailed}	r	p _{one-tailed}
Left vmPFC volume at baseline	-0.194	0.035	-0.126	0.048
Right vmPFC Development	0.118	0.157	-0.124	0.050

Correlations between brain features and smoking (among females)

Brain Features	Lifetime occasions of smoking at Baseline among BL-S Group (n = 97)		Lifetime occasions of smoking at follow-up among FU-S Group (n = 193)	
	r	p one-tailed	r	p one-tailed
	Left vmPFC volume at baseline	-0.166	0.048	-0.102
Right vmPFC Development	0.068	0.263	-0.133	0.032

(5) Some of the methods are also unclear, for example, p.18, line 430 'only white men were included' versus p.18 line 435 in which sex is listed as a covariate.

Reply: We are very sorry for the typo, and the sentence has now been revised to clarify that we included white people (both males and females) in the study.

*"To ensure homogeneity of the datasets, only **white people** were included to validate the relationship between the GMV of left vmPFC and rule-breaking behaviour and discover the genetic factors underlying the neurobehavioural circuit."*

(6) The Discussion lacks acknowledgement of clear limitations, including absence of sex comparisons and the overall relatively small effect sizes for most analyses.

Reply: We have added a limitation statement in Discussion as indicated in our reply to your first question.

"It is a limitation that the current study mainly focused on the initiation and the early stage of addictive behaviour, but the onset of actual addiction requires possibly aberrant reinforcement processes operating long-term⁵⁴. Nevertheless, we did manage to validate the proposed differentiated roles of the left and right vmPFC GMV in addictive behaviour (i.e., with initiation and sustenance/dependence respectively) in daily smokers from the independent HCP data. In future studies, it would be of considerable interest to understand how sustained smoking behaviour, driven initially by hedonic experience may further develop into dependence (i.e., psychological or physical craving for nicotine following withdraw), where habit-inducing regions such as the insula and striatum might eventually become involved^{55,56}. Therefore, longitudinal data of patients with substance use disorder and relevant animal models are crucial for the future studies. The other limitation of this study was that some of the genetic-behavioural findings in the generalisation cohort were relatively small, which could be largely due to the fact that the genetic constructs of left vmPFC GMV were highly time sensitive (Table S8 and Fig. S7). Furthermore, in the context of the vaping epidemic among adolescents, whether there are similar neurobehavioural mechanisms between neurodevelopment and e-cigarette use needs further investigation."

Reviewer #3 (Remarks to the Author):

The current manuscript from Xiang et al. describes an analysis of smoking behaviors and gray matter volume in the longitudinal IMAGEN sample (assessed at ages 14, 19, and 23, though most analyses reported are comparisons across baseline (age 14) and the first follow-up (age 19)). The authors were interested in testing whether there are structural brain differences that predispose one to initiate smoking, and/or whether sustained smoking behavior results in structural brain differences. They find evidence that reduced gray matter volume (GMV) in the left vmPMC is potentially causal for smoking initiation (via rule-breaking). Conversely, they find that sustained smoking behavior may be causal for reduction in GMV in the right vmPFC.

This article addresses important questions about the relationships between smoking and brain development, leveraging a longitudinal sample with genetic, neuroimaging, and smoking (and other behavioral) measures to address causality. This is a real strength of the study, and I think the results from this study will be of great interest to the field. However, I have several suggestions for improvement, detailed below (Note: I am not an expert in neuroimaging approaches, so I have not commented as much on those aspects of the paper, but have mainly focused on the genetic analyses.)

Comments:

(1) The genetic analyses include GWAS of GMV, rule-breaking, and smoking quantity in subjects of the IMAGEN study who were not included in the other analyses due to lack of neuroimaging data at follow-up (N = 1,026). "Valid PRS" were then scored in the analytic sample of IMAGEN (N = 752) using these GWAS results, and used as indicators in a Mendelian Randomization approach. Could the authors please provide more analysis details about these GWAS? For example, did the authors generally follow the same quality control procedures as they did for the larger GWAS meta-analysis of GMV of the left vmPFC? What covariates were included in the GWAS? Did the authors ensure that none of the individuals in the "discovery" GWAS (N = 1,026) were genetically related to those in the "target" PRS sample (N = 752)?

Reply: We have revised the procedures for GWAS analysis in Methods as follows:

"Specifically, using PLINK ⁶⁶, we conducted exploratory GWAS for the phenotypes of interest in the leftover participants excluded from the above analyses due to the lack of neuroimaging information at FU (n = 1,026) in the IMAGEN project. We first performed the quality-control processing using PLINK ⁶⁶, where SNPs with call rates <95%, minor allele frequency <0.1%, deviation from the Hardy-Weinberg equilibrium with p <1E-10 were excluded from the analysis. Then we conducted the imputation on the quality-controlled genetic data with TOPMed imputation server (<https://imputation.biodatacatalyst.nih.gov>). After imputation, 5,966,316 SNPs were available for IMAGEN sample. The following GWAS analysis was performed with sex, research sites and top 10 ancestry principal components as covariates. (see Fig. S8 for the plot of the first two PCs projected to 1000 Genome data)."

The same quality control procedure was also used for the ABCD data, which is now updated in

Methods:

"To discover the genetic factors underlying the neurobehavioural circuit from the left vmPFC to rule-breaking behaviour, we performed a meta-analysis of GWAS for GMV of the left vmPFC. For the ABCD cohort, we performed the same quality control and imputation procedures as the above IMAGEN study. After imputation, 4,244,228 SNPs of 4,390 participants (with complete phenotypes and no siblings) from ABCD were available. GWAS was performed using PLINK with sex, research sites and top 20 ancestry principal components as covariates (see Fig. S8 for the plot of the first two PCs projected to 1000 Genome data)."

Regarding the co-ancestry level, we estimated the IBD between all IMAGEN participants (with PLINK 2.0), and all indices were much less than 0.125, the commonly used removal threshold (i.e. third-degree relatives). Therefore, no participant was removed due to genetic relationship. This information has now been added in the relevant section of Method:

"PRSs were hence calculated with clumped SNPs for individuals with complete longitudinal neuroimaging, behaviour and genetic information ($n = 752$) at a pre-defined set of P-value thresholds (i.e., 0.05, 0.1, 0.2, 0.3, 0.4 and 0.5). Please be noted that the highest co-ancestry level (i.e., the proportion of shared genome estimated using PLINK 2.0) in the IMAGEN cohort is much lower than the common threshold 0.125, i.e., third-degree relatives, for exclusion (Fig. S9), so the risk for information leakage due to relatedness between the discovery GWAS sample and the target PRS sample is neglectable."

Fig. S9 The pairwise co-ancestry levels (i.e., the proportion of shared genome estimated using PLINK 2.0) between participants from the IMAGEN cohort.

(2) The authors also conducted a separate GWAS meta-analysis of GMV of the left vmPFC in the full IMAGEN cohort ($N = 1,778$) and the ABCD cohort ($N = 4,390$). However, the top SNP in their analysis did not replicate in the UK Biobank cohort ($N = 38,677$), and neither was the PRS associated in the UK Biobank. The authors speculate time-specific genetic effects on left vmPFC GMV, and while that certainly might be true, that this is evidence for couldn't this also potentially represent a false positive in

the authors' GWAS?

Reply: To confirm whether the identified genetic effects are indeed time-specific, we validated the polygenic risk score (PRS) of left vmPFC GMV in different datasets with varied age ranges and observed gradually diminished effect sizes with increased age gaps.

To elaborate, the original GWAS was conducted in an IMAGEN sub-sample without longitudinal data (N = 1,026) at age 14, and we first evaluated the hence established PRS in an independent IMAGEN sub-sample with longitudinal data (N = 737) and found that the PRS could significantly predict, though with gradually reduced effects, left vmPFC GMV across all three timepoints. Further, we investigated the predictive performance of this PRS in other datasets representing different age ranges (for instance, ABCD, ages 9-10; HCP, ages 22-37; UKB1, ages 40-50; UKB2, ages 50-60; UKB3, ages > 60 years, as summarised in Table S8 below).

Table S8. Correlations with the PRS of left vmPFC in different datasets

Dataset	Age	Age Gap	Sample Size	r_{mean}	$p_{one-tailed}$
IMAGEN-BL*	14	0	737	0.393	< 0.001
ABCD	9-10	~4.5	4,390	0.141	< 0.001
IMAGEN-FU	19	5	737	0.146	< 0.001
IMAGEN-FU2	23	9	737	0.112	< 0.001
HCP	22-37	~16	672	0.070	0.04
UKB1	40-50	~30	19,346	0.011	0.06
UKB2	50-60	~40	16,596	0.006	0.22
UKB3	> 60	~50	10,145	0.004	0.34

*The same age as the discovery sample.

With all the above data (i.e., three timepoints from the IMAGEN, one timepoint from the ABCD, one timepoint from the HCP and three timepoints from the UKB), a Spearman's ranked test would indicate that the absolute age gap demonstrated an almost perfect negative correlation $\rho = -0.976$ with the predictive effects of PRS ($p < 0.001$), i.e. a monotonic decreasing. More precisely, the predictive effects of PRS reduced exponentially with increased age gaps ($r = -0.985$, $p < 0.001$ for age gaps and $\log(\text{predictive effects})$), hence strongly indicating an age-specific genetic effect for the vmPFC GMV. We have now revised the relevant sections in the manuscript accordingly:

“Notably, evidence also indicated that the genetic factors of GMV in the left vmPFC might be time sensitive for a specific development period adolescence, as the corresponding PRS demonstrated a series of rapidly diminished predictive effects across cohorts with increased age gaps deviated from age 14 (i.e., IMAGEN-BL, age 14; ABCD, ages 9-10; IMAGEN-FU, age 19; IMAGEN-FU2, age 23; HCP, ages 22-37; UKB1, ages 40-50; UKB2, ages 50-60; UKB3, ages > 60 years; Spearman’s ranked test: $\rho = -0.976$, $p < 0.001$; $r = 0.985$, $p < 0.001$ for a log-regression model; Table S8 and Fig. S7).”

Fig. S7 The predictive effects of PRS (i.e. the correlation r) reduced exponentially (plot as $\log(r)$) with increased age gaps.

(3) I believe neuroimaging phenotypes show slightly larger genetic effect sizes on average, compared to behavioral traits, but the GWAS N (meta-analyzed = 6,168) still seems relatively low-powered for this analysis. It would be helpful if the authors could provide a sensitivity analysis that estimates how much statistical power they had to detect reasonable effect sizes based on previous literature.

Reply: Previous GWAS often identified top SNPs of brain grey matters with a correlation over 0.1 (or equivalently variance explained > 1.0%) (Grasby et al., 2020; Luo et al., 2019). Therefore, with a predefined effect size $r = 0.1$, 6000 individuals are sufficient (with a statistical power = 0.99) to identify this signal with a false positive rate at $5.0E-8$, i.e. the genome-wide significant level, for a two-tailed test.

References:

Grasby, K. L., Jahanshad, N., Painter, J. N., Colodro-Conde, L., Bralten, J., Hibar, D. P., . . . Medland, S. E. (2020). "The genetic architecture of the human cerebral cortex." *Science* 367(6484), eaay6690.

Luo, Q., Chen, Q., Wang, W., Desrivières, S., Quinlan, E. B., Jia, T., . . . consortium, I. (2019). "Association of a Schizophrenia-Risk Nonsynonymous Variant With Putamen Volume in Adolescents: A Voxelwise and Genome-Wide Association Study." *JAMA Psychiatry*, 76(4), 435-445.

(4) Relatedly, it might be helpful for the authors to present the effect sizes of the top SNP(s) in the IMAGEN and ABCD cohorts, either in a forest plot or a supplementary table, so readers can assess whether the signal is being driven primarily by one cohort or by both.

Reply: The correlations between the top SNP and the left vmPFC GMV were -0.08 ($p = 1.6E-7$) and -0.11 ($p = 1.2E-5$) in ABCD and IMAGEN cohorts, respectively. We further performed a meta-analysis across the research sites for each cohort (see the forest plots in Fig. S6 below). The effect sizes from different sites were highly consistent in both cohorts, and the meta-analyses results

were similar to the original analyses for both ABCD ($Z = -5.46$ $p = 4.7E-8$) and IMAGEN ($Z = -4.27$ $p = 1.9E-5$).

Fig. S6 The forest plots for meta-analyses of the lead SNP rs17699090's associations with gray matter volume of the left vmPFC across research sites in both (a) ABCD and (b) IMAGEN cohorts.

(5) Given the relatively low statistical power of their samples, I would opt not to include in the text the associations of the 'G' allele at the top SNP with rule-breaking, conduct problems, or smoking behavior, especially as these p-values don't seem to be corrected for any multiple testing and range from 0.02 to 0.049.

Reply: Given the low frequency of the risk allele G of rs17699090 (MAF=0.088), we re-investigated this SNP's associations with behaviours focusing on a contrast of G-allele carriers vs the non-carriers. Consistent with our proposed causal link "left vmPFC -> rule breaking -> smoking", G-allele carriers have significantly higher rule breaking scores than the non-carriers in both the IMAGEN ($t_{1388} = 2.45$, Cohen's $d = 0.13$, $p_{one-tailed} = 0.007$) and ABCD cohorts ($t_{4264} = 3.31$, Cohen's $d = 0.10$, $p_{one-tailed} < 0.001$, Fig. 3d). Both results could survive the FDR correction. Further, smokers in the IMAGEN cohort did carry more G-alleles of rs17699090 than those in controls ($t_{725} = 2.12$, Cohen's $d = 0.12$, $p_{one-tailed} = 0.018$).

"Additionally, complying with the proposed causal effects of lower GMV in the left vmPFC identified above, the participants with the G-allele of rs17699090 have higher rule-breaking score at follow-up interview than those without the G-allele in both the IMAGEN ($t_{1388} = 2.45$, Cohen's $d = 0.13$, $p_{one-tailed} = 0.007$) and ABCD cohorts ($t_{4264} = 3.31$, Cohen's $d = 0.10$, $p_{one-tailed} < 0.001$, Fig. 3d). Further, smokers in the IMAGEN cohort did carry more G-alleles of rs17699090 than those in controls ($t_{725} = 2.12$, Cohen's $d = 0.12$, $p_{one-tailed} = 0.018$)."

The other analyses, such as the associations with conduct problems, were now removed in the revised text as suggested.

(6) The authors' creation of "valid PRS" seems appropriate, although it would be helpful if more information was provided on this method – has this exact approach been used before? (The authors

reference a few MR papers, including Burgess et al. (2020), but it does not appear that the Burgess et al. paper used this PRS approach.) If the authors are introducing a new MR method, it might increase readers' confidence in this new approach if the authors were to demonstrate its feasibility first with well-established causal relationships (e.g., LDL cholesterol and heart disease), before applying their method to the traits of interest (GMV of vmPFC, smoking behaviors, sensation-seeking).

Reply: This approach has been first introduced in our previous study (Kang, Jia et al. 2022a). We have now cited it in the Results and Methods section. Also, in a recent preprint (Kang, Jia et al. 2022b), we utilised several established Mendelian randomisation (MR) methods along with this PRS approach to investigate the potential causality of the body mass index and brain structure. The causal inference from this newly developed approach was consistent with those obtained from the other established MR methods (Kang, Jia et al. 2022b).

References:

- Kang, J., T. Jia, Z. Jiao, C. Shen, C. Xie, W. Cheng, B. J. Sahakian, D. Waxman and J. Feng (2022a). "Increased brain volume from higher cereal and lower coffee intake: shared genetic determinants and impacts on cognition and metabolism." *Cerebral Cortex*.
- Kang, J., T. Jia, Z. Linli, Y. Li, W. Cheng, S. Guo and J. Feng (2022b). "Association between obesity, brain atrophy and accelerated brain aging and their genetic mechanisms." 2022.2012.2030.22284052.

(7) In the Methods section, the authors describe the ABCD Study analytic sample (for the GWAS meta-analysis) as being confined to "white men" and say that they focused on "white people from both IMAGEN and ABCD cohorts". First, given the age of the sample, it might be more accurate to describe the ABCD sample as consisting of white males (instead of men), or male children whose parents reported their child's race as being "white".

Reply: Our sincere apologies for the typo, and we have revised this sentence to clarify that we have used all the white people, including both males and females, in the ABCD dataset.

(8) Second, I assume the authors are using self-reported race here, but it is more typical for genetic analyses to use genetic principal components to define more homogeneous genetic ancestry groups without the use of self-reported race, or to "confirm" self-reported race using genetic principal components (i.e., cross-check self-reported race with clustering on genetic PCs). Could the authors comment on their choice of approach here? It might also be helpful to see a PCA plot of their analytic samples, with the PCs projected onto the major 1000 Genomes ancestry groups. Furthermore, the authors don't appear to have corrected for genetic PCs in the IMAGEN study – yet even in a "homogeneous" population, correcting for PCs is important to control for genetic structure (<https://doi.org/10.1038/nature07331>).

Reply: We indeed used self-reported races to select our samples and then controlled for potential population stratification with top 20 ancestry PCs in ABCD (in IMAGEN, we controlled for 10 PCs). We are sorry for the confusion, and the relevant section in Methods has been revised to as follows:

For IMAGEN:

"The following GWAS was performed with sex, research sites and top 10 ancestry principal components as covariates."

For ABCD:

"GWAS was performed using PLINK with sex, research sites and top 20 ancestry principal components as covariates."

As suggested by the reviewer, we now provide PCA plots for both IMAGEN and ABCD samples with their ancestry PCs projected onto the major 1000 Genomes ancestry groups (see Fig. S8 attached below). Notably, for both IMAGEN and ABCD cohorts, most participants were Europeans, with a small proportion of admixed Americans (i.e. with European backgrounds). Overall, the self-rated races of being 'white' are highly accurate (IMAGEN participants were considered white according to the enrolment criteria (Mascarell Maricic, Walter et al. 2020), and regressing out top 10-20 ancestry PCs should be sufficient to control for any cryptic population stratification.

Fig. S8 Plots of first two components of IMAGEN (left) and ABCD (right) projected to 1000 Genome ancestry groups.

References:

Mascarell Maricic, L., Walter, H., Rosenthal, A., Ripke, S., Quinlan, E. B., Banaschewski, T., . . . consortium, I. (2020). "The IMAGEN study: a decade of imaging genetics in adolescents." *Mol Psychiatry*, 25(11), 2648-2671.

(9) It was unclear to me whether the authors tested associations only with gray matter volume, or whether they tested other modalities as well. If they did not test other modalities, I think it would be helpful for the authors to provide a bit more explanation of their study motivations/hypotheses in the Introduction and justify their choice to focus only on gray matter volume.

Reply: We only focused on gray matter volume (GMV) but no other MRI modalities. This choice is rather intuitive as GMV (a measurement directly related to neuronal cell bodies, i.e., the primary components of the gray matter in the brain) and its morphological features have so far been the most commonly used neuroimaging measures for studying neurodevelopment. On the other hand, neurodevelopment during adolescents is a critical transition period, and brain maldevelopment

(as either a cause or an outcome) has been implicated in addictive behaviour such as smoking. For instance, many studies on the relationship between addictive behaviours and neurodevelopment have focused explicitly on GMV (Gray, Thompson et al. 2020, Li, Liu et al. 2020, Robert, Luo et al. 2020, Albaugh, Ottino-Gonzalez et al. 2021, Conti and Baldacchino 2021). We have now extended the second section of Introduction to highlight the importance of neurodevelopment for addictive behaviour as follows:

*"Several previous studies have attributed smoking initiation to impaired executive control and the underlying neural circuits^{13,21}. For instance, the prefrontal cortex (PFC), the most critical neural network engaged in response inhibition and risk adjustment, continues to develop structurally and functionally into adulthood^{20,22}. **Its disrupted development has been implicated as a trigger for maladaptive behaviour, such as addiction^{19,20,23}. On the other hand, substance use, including nicotine exposure, may cause damage to the brain and accelerate brain aging potentially through its neurotoxic properties, indirectly exacerbated by excessive smoking^{8,10}. Such neurotoxic effects may also affect the reinforcement system itself and induce other forms of substance dependence^{24,25}."***

References:

Albaugh, M. D., J. Ottino-Gonzalez, A. Sidwell, C. Lepage, A. Juliano, M. M. Owens, B. Chaarani, P. Spechler, N. Fontaine, P. Rioux, L. Lewis, S. Jeon, A. Evans, D. D'Souza, R. Radhakrishnan, T. Banaschewski, A. L. W. Bokde, E. B. Quinlan, P. Conrod, S. Desrivieres, H. Flor, A. Grigis, P. Gowland, A. Heinz, B. Ittermann, J. L. Martinot, M. L. Paillere Martinot, F. Nees, D. Papadopoulos Orfanos, T. Paus, L. Poustka, S. Millenet, J. H. Frohner, M. N. Smolka, H. Walter, R. Whelan, G. Schumann, A. Potter, H. Garavan and I. Consortium (2021). "Association of Cannabis Use During Adolescence With Neurodevelopment." JAMA Psychiatry.

Conti, A. A. and A. M. Baldacchino (2021). "Neuroanatomical Correlates of Impulsive Choices and Risky Decision Making in Young Chronic Tobacco Smokers: A Voxel-Based Morphometry Study." Front Psychiatry 12: 708925.

Gray, J. C., M. Thompson, C. Bachman, M. M. Owens, M. Murphy and R. Palmer (2020). "Associations of cigarette smoking with gray and white matter in the UK Biobank." Neuropsychopharmacology 45(7): 1215-1222.

Li, J., B. Liu, T. Banaschewski, A. L. W. Bokde, E. B. Quinlan, S. Desrivieres, H. Flor, V. Frouin, H. Garavan, P. Gowland, A. Heinz, B. Ittermann, J. L. Martinot, E. Artiges, F. Nees, D. Papadopoulos Orfanos, T. Paus, L. Poustka, S. Hohmann, J. H. Frohner, M. N. Smolka, H. Walter, R. Whelan, G. Schumann, I. Consortium and T. Jiang (2020). "Orbitofrontal cortex volume links polygenic risk for smoking with tobacco use in healthy adolescents." Psychol Med: 1-8.

Robert, G. H., Q. Luo, T. Yu, C. Chu, A. Ing, T. Jia, D. Papadopoulos Orfanos, E. Burke-Quinlan, S. Desrivieres, B. Ruggeri, P. Spechler, B. Chaarani, N. Tay, T. Banaschewski, A. L. W. Bokde, U. Bromberg, H. Flor, V. Frouin, P. Gowland, A. Heinz, B. Ittermann, J. L. Martinot, M. L. Paillere Martinot, F. Nees, L. Poustka, M. N. Smolka, N. C. Vetter, H. Walter, R. Whelan, P. Conrod, T. Barker, H. Garavan, G. Schumann and I. Consortium (2020). "Association of Gray Matter and Personality Development With Increased Drunkenness Frequency During Adolescence." JAMA Psychiatry 77(4): 409-419.

(10) Similarly, it was unclear to what extent the authors tested multiple regions and only saw significant effects in the left vmPFC and right vmPFC, and thus only reported on those effects, or whether they chose *a priori* to focus on these regions. This should be more clearly spelled out in the manuscript.

Reply: We are sorry for the confusion here. When comparing both smoker groups (i.e., the BL-Smoker and the FU-Smoker) with the control group, we investigated the brain-wide associations, instead of focusing on the frontal area only.

Indeed, in the second paragraph of Results, apart from both left and right vmPFC, we have reported several other brain regions, all with significantly reduced GMV at baseline (corrected for brain-wide multiple-comparison using the family-wise error rate, FWE), such as the left inferior frontal cortex and the left lateral orbitofrontal cortex, when comparing BL-Smoker with the controls (Fig. 1a). However, in the following comparison between the FU-Smoker and the controls (Fig. 1a & 2a), only the left vmPFC measured at the baseline was found significantly smaller (FWE-adjusted), i.e., prior to the actual initiation of smoking. Therefore, we proposed that reduced GMV in the left vmPFC could be a highly sensitive risk factor for the initiation of future smoking behaviour and focused on this region henceforward.

We have now revised the relevant sections to make it clearer that a brain-wide approach was adopted:

*“At baseline, compared with the controls, **a whole brain analysis found BL-S had significantly smaller gray matter volumes (GMV) in clusters such as the ventromedial prefrontal cortex (vmPFC)/anterior cingulate cortex (ACC) (Peak Montreal Neurological Institute (MNI): [-2, 44, 6] Brodmann Area [BA] 10_L, $t_{426} = 4.46$, Cohen’s $d = 0.44$, Cluster: 3962 voxels, $p_{FWE-adj} < 1E-6$), the left inferior frontal cortex (IFC, Peak MNI: [-50, 26, 27] BA 48_L, $t_{426} = 4.23$, Cohen’s $d = 0.40$, Cluster: 446 voxels, $p_{FWE-adj} = 0.009$), the left lateral orbitofrontal cortex (latOFC, Peak MNI: [-36, 47, -3] BA 47_L, $t_{426} = 4.21$, Cohen’s $d = 0.40$, Cluster: 775 voxels, $p_{FWE-adj} = 0.001$) and the right dorsolateral prefrontal cortex (dlPFC, Peak MNI: [39, 15, 27] BA 48_R, $t_{426} = 4.03$, Cohen’s $d = 0.40$, Cluster: 796 voxels, $p_{FWE-adj} = 0.001$) (Fig. 1a upper). Remarkably, FU-S also had smaller GMV than the controls at baseline, i.e., prior to their smoking initiation, **only in the left vmPFC from a whole-brain analysis (Peak MNI: [-5, 50, -5] BA 10_L, $t_{611} = 4.19$, Cohen’s $d = 0.34$, Cluster: 438 voxels, $p_{FWE-adj} = 0.011$) (Fig. 1a lower), of which 426 voxels overlapped with the vmPFC cluster differentiating BL-S from the controls (Fig. 1b upper).”*****

Similarly, when investigating how the longitudinal trajectory across the whole brain could differentiate FU-smokers from the controls (a putative outcome of smoking), we only identified a significant area in the right vmPFC (FWE-adjusted). We have also revised the relevant section to clarify that a brain-wide approach was adopted:

*“We next investigated if smoking could affect the development of GMV (Fig. S1, see Methods for more details) and observed faster GMV reduction (i.e., from baseline to follow-up) in the right vmPFC **only** (Peak MNI: [10, 41, -11] BA 10_R, $t_{609} = -4.16$, Cohen’s $d = -0.34$, Cluster: 747 voxels, $p_{FWE-adj} = 0.008$) when comparing FU-S to the controls **with a whole brain analysis** (Fig. 2a).”*

Therefore, the focus on the left and right vmPFC in the present study was purely data-driven.

Minor comments:

(1) The mediation model shown in Extended Data Figure 2 is a bit confusing to me – it seems to show follow-up smoking behavior being the mediator of a relationship between left vmPFC at baseline and sensation seeking at follow-up, but this seems counterintuitive – wouldn't we expect sensation seeking to be the mediator of a relationship between left vmPFC and smoking?

Reply: According to the between-group comparisons and within-group association analyses, our results indicated that the reduced GMV in left vmPFC preceded future smoking initiation (i.e., a potential causal effect). However, while the baseline left vmPFC GMV was associated with the quantity of smoking at both baseline and follow-up, its association with sensation seeking was only observed at follow-up. Thus, we expect smoking to mediate the relationship between the baseline left vmPFC GMV and sensation seeking at follow-up.

As a complementary analysis to the original analysis, we also performed an additional mediation analysis with sensation seeking as the mediator and found no significant mediation effect (see the revised Fig. S2 below).

In a later section, “R-vmPFC’s enhanced role in reward-seeking”, we hypothesised that sensation seeking requires an accumulation of hedonic experience to reinforce addictive behaviour further. Thus, with the proposed causal relationship of the left vmPFC to smoking (via rule-breaking behaviour but not sensation seeking), it is not surprising to find sensation seeking coming secondary to the smoking experience in the above mediation analyses.

(2) The abbreviation "MNI" is introduced in the Results section with no explanation (e.g., "Peak MNI: [-2, 44, 6]") – could the authors provide the full version of this abbreviation at first mention?

Reply: Revised as suggested.

(3) Similarly, Table 1 could use explanations for all of the abbreviations (e.g., TIV, BMI, WISCIV).

Reply: Revised as suggested.

(4) For several of the Extended Data Tables, the p-values are given as "0.000". Could the authors please present a p-value in scientific notation here?

Reply: Revised as suggested.

(5) Inconsistent scientific notation: at one place in the text, a p-value is given as "1E-3", whereas other times similar p-values are given as < 0.001 .

Reply: We have unified all the notations for the p-values.

Reviewer #1 (Remarks to the Author):

The authors have addressed my comments thoroughly and I do not have any other concerns. I consider this manuscript acceptable for publication.

Reviewer #2 (Remarks to the Author):

The authors have addressed my concerns

Reviewer #3 (Remarks to the Author):

The authors have thoroughly addressed all of my questions in their response letter.

I only have one remaining suggestion, which is to add "self-report" to the Methods description of the ABCD GWAS sample (lines 512-515 of the PDF): "To ensure homogeneity of the datasets, only [self-reported] white people were included to validate the relationship between the GMV of left vmPFC and rule-breaking behaviour and discover the genetic factors underlying the neurobehavioural circuit". I think it's especially important in studies involving genetic analyses to be clear about how sample groups are determined, especially given the NASEM's recent guidelines on population descriptors in genomics research:
<https://nap.nationalacademies.org/catalog/26902/using-population-descriptors-in-genetics-and-genomics-research-a-new>

REVIEWER COMMENTS

Reviewer #1 (Remarks to the Author):

The authors have addressed my comments thoroughly and I do not have any other concerns.
I consider this manuscript acceptable for publication.

Reply: We thank the reviewer for the acknowledgement.

Reviewer #2 (Remarks to the Author):

The authors have addressed my concerns

Reply: We thank the reviewer for the acknowledgement.

Reviewer #3 (Remarks to the Author):

The authors have thoroughly addressed all of my questions in their response letter.

I only have one remaining suggestion, which is to add "self-report" to the Methods description of the ABCD GWAS sample (lines 512-515 of the PDF): "To ensure homogeneity of the datasets, only [self-reported] white people were included to validate the relationship between the GMV of left vmPFC and rule-breaking behaviour and discover the genetic factors underlying the neurobehavioural circuit". I think it's especially important in studies involving genetic analyses to be clear about how sample groups are determined, especially given the NASEM's recent guidelines on population descriptors in genomics research: <https://nap.nationalacademies.org/catalog/26902/using-population-descriptors-in-genetics-and-genomics-research-a-new>

Reply: We thank the reviewer for the helpful suggestion and have revised the relevant sentence accordingly.

*"...To ensure homogeneity of the datasets, only **self-reported** white people were included to validate the relationship between the GMV of left vmPFC and rule-breaking behaviour and discover the genetic factors underlying the neurobehavioural circuit (see Fig. S8 for the plot of the first two PCs projected to 1000 Genome data) ..."*